# Two chemically distinct root lignin barriers control solute and water balance

Guilhem Reyt[1], Priya Ramakrishna [1,10], Isai Salas-González [2], Satoshi Fujita [3,11], Ashley Love[4],
David Tiemessen[4], Catherine Lapierre[5], Kris Morreel[6,7], Monica Calvo-Polanco [8,12], Paulina Flis [1],
Niko Geldner [3], Yann Boursiac [8], Wout Boerjan [6,7], Michael W. George[4,9], Gabriel Castrillo[1] &
David E. Salt [1✉]

Lignin is a complex polymer deposited in the cell wall of specialised plant cells, where it provides essential cellular functions. Plants coordinate timing, location, abundance and composition of lignin deposition in response to endogenous and exogenous cues. In roots, a fine band of lignin, the Casparian strip encircles endodermal cells. This forms an extracellular barrier to solutes and water and plays a critical role in maintaining nutrient homeostasis. A signalling pathway senses the integrity of this diffusion barrier and can induce over-lignification to compensate for barrier defects. Here, we report that activation of this endodermal sensing mechanism triggers a transcriptional reprogramming strongly inducing the phenylpropanoid pathway and immune signaling. This leads to deposition of compensatory lignin that is chemically distinct from Casparian strip lignin. We also report that a complete loss of endodermal lignification drastically impacts mineral nutrients homeostasis and plant growth.

[1] Future Food Beacon of Excellence & School of Biosciences, University of Nottingham, Sutton Bonington, UK. [2] Curriculum in Bioinformatics and Computational Biology, Department of Biology, University of North Carolina at Chapel Hill, Chapel Hill, NC, USA. [3] Department of Plant Molecular Biology, Biophore, University of Lausanne, Lausanne, Switzerland. [4] School of Chemistry, University of Nottingham, Nottingham, UK. [5] Institut Jean-Pierre Bourgin, INRAE, AgroParisTech, Université Paris-Saclay, Versailles, France. [6] Department of Plant Biotechnology and Bioinformatics, Ghent University, Ghent, Belgium. [7] Center for Plant Systems Biology, VIB, Ghent, Belgium. [8] Biochimie & Physiologie Moléculaire des Plantes, University of Montpellier, CNRS, INRAE, SupAgro, Montpellier, France. [9] Department of Chemical and Environmental Engineering, The University of Nottingham Ningbo China, Ningbo, China. [10]Present address: Department of Botany and Plant Biology, University of Geneva, Geneva, Switzerland. [11]Present address: National Institute of Genetics, Mishima, Shizuoka, Japan. [12]Present address: Excellence Unit AGRIENVIRONMENT, CIALE, University of Salamanca, Salamanca, Spain.
✉email: david.salt@nottingham.ac.uk

Lignin is a phenolic polymer and is one of the main components of secondary-thickened cell wall (CW) in vascular plants. Its chemical properties give strength, stiffness and hydrophobicity to the CW. Lignin provides mechanical support, modulates the transport of water and solutes through the vascular systems, and provides protection against pathogens[1,2]. Lignin polymerization occurs through oxidative coupling of monolignols and other aromatic monomers[3,4]. The monolignols that is *p*-coumaryl, coniferyl and sinapyl alcohols are synthesized from the amino acid phenylalanine through the phenylpropanoid pathway. They are then polymerized into lignin to form the *p*-hydroxyphenyl (H), guaiacyl (G) and syringyl (S) subunits of the lignin polymer. Lignin composition and abundance are highly variable among and within plants species, tissues and cell types and can be modulated by environmental cues[1].

In roots, large amounts of lignin are deposited in xylem vessels, an important component of the vascular system[5,6]. Lignin is also deposited in the endodermal cells surrounding the vascular tissues, for Casparian strip (CS) formation[7]. Both the vascular system and the CS play a critical role in water and mineral nutrient uptake from the soil, and their transport toward the shoot[8–10]. In *Arabidopsis thaliana*, the composition of lignin monomers in the CS and xylem is similar with a strong predominance of the G-unit (>90%)[7]. However, the machinery required for CS lignification appears to be distinct from that needed for xylem lignification[6,11].

The deposition of the CS in the endodermal CW prevents the apoplastic diffusion of solutes between the outer and inner tissues of the root, forcing solutes to pass through the symplast of endodermal cells[8]. CS lignin encircles each endodermal cell, forming a bridge between them. This precise lignin deposition is defined by the presence of the transmembrane Casparian strip membrane domain proteins (CASPs)[12], peroxidases[13,14] and the dirigent-like protein ESB1[15]. The expression of this lignin polymerization machinery is tightly controlled by the transcription factor MYB36[16,17]. A surveillance mechanism for CS integrity, called the Schengen-pathway, boosts CS deposition and is necessary for CS fusion and sealing of the extracellular space (apoplast)[18]. This pathway involves vasculature-derived peptides CASPARIAN STRIP INTEGRITY FACTORS 1 and 2 (CIF1 and 2)[19,20] and their perception by the leucine-rich repeat receptor-like kinase SCHENGEN3 (SGN3, also called GSO1). Their interaction triggers a cascade of signalling events mediated by kinases, that involves SGN1, and the activation of the NADPH oxidase RBOHF (SGN4) leading to the ROS production necessary for lignin polymerization[13,18,21]. These kinase signalling events occur on the cortex-facing side of the CS and mediate the transition from a discontinuous CS with islands of lignin into a continuous CS with its characteristic ring of lignin sealing the apoplast[18]. Once the CS is sealed, CIF peptide diffusion is blocked and the Schengen-pathway becomes inactive. In mutants with an impaired CS, such as *esb1* and *myb36*, the Schengen-pathway is constitutively activated due to a constant leak of the CIF(s) peptides through the CS[12,15,17,18,22]. This induces in endodermal CW compensatory lignification in the cell corners and suberisation of the cell surface. However, the role of this compensatory lignin, and the mechanism controlling its deposition are not fully understood.

Here, we demonstrate that constitutive activation of the Schengen-pathway induces the deposition of compensatory lignin in the corners of endodermal cells that is chemically distinct from CS lignin. We characterized this lignin and found commonalities with stress- and pathogen-response lignin, which has a high content of the H subunit. Furthermore, we demonstrate that this cell-corner lignification is preceded by a transcriptional reprogramming of endodermal cells, causing a strong induction of the

phenylpropanoid pathway, and a significant inactivation of aquaporin expression. Our findings also establish that the activation of the Schengen-pathway, in order to compensate for a defective CS, is of critical importance for plants to maintain their mineral nutrients homoeostasis and water balance.

## Results and discussion

**Two pathways of endodermal lignification.** In order to disentangle the role of MYB36 and SGN3 in controlling endodermal lignification, we generated the double mutant *sgn3-3 myb36-2*. We analyzed the endodermal accumulation of lignin in *sgn3-3 myb36-2*, and the corresponding single mutants *sgn3-3* and *myb36-2* (Fig. 1a–c). In the early stage of endodermal differentiation, six cells after the onset of elongation, we observed deposition of CS lignin in "a string of pearl" manner in WT and *sgn3-3* (Fig. 1a). No endodermal lignification was observed in *myb36-2* or *sgn3-3 myb36-2* at this early developmental stage (Fig. 1a). Later in endodermal development, ten cells after the onset of elongation, we observed a continuous CS ring of lignin, sealing the endodermal cells in WT plants (Fig. 1a–c). As we expected, in *sgn3-3* impaired in the activation of the Schengen-pathway, CS lignification is discontinuous[9], while *myb36-2* exhibits compensatory lignification in the corners of endodermal cells facing the cortical side of the endodermis, as previously reported[9,17]. In contrast, no lignification at the CS or cell corners was observed in *sgn3-3 myb36-2* (Fig. 1a–c). These results establish that the cell-corner compensatory lignification observed in *myb36-2* lacking a CS (17) is SGN3 dependent.

To test how these different patterns of endodermal lignification found in WT, *sgn3-3*, *myb36-2* and *sgn3-3 myb36-2* affect the permeability of the root apoplast, we assessed the penetration into the stele of the fluorescent apoplastic tracer propidium iodide (PI)[23] (Fig. 1d). We quantified the percentage of the root length permeable to PI and found it partially increased in both *sgn3-3* and *myb36-2*, in comparison to WT (Fig. 1d). Surprisingly, we observed in *sgn3-3 myb36-2* that the entire length of the root was permeable to PI, indicating an additive effect of *sgn3-3* and *myb36-2* mutations. This result suggests that MYB36 and SGN3 control apoplastic sealing through two independent lignification pathways. The lack of compensatory cell-corner lignification in *sgn3-3 myb36-2* could explain the complete permeability of the roots in the double mutant. This finding supports recent observations assigning a role as an apoplastic barrier to SGN3-dependent cell-corner lignification[14]. In addition, constitutive activation of the Schengen-pathway is also known to trigger an enhanced suberisation in certain CS mutants, including *myb36*[17]. We confirm this observation in *myb36-2* (Fig. 1e). This enhanced suberisation in *myb36* is also SGN3 dependent since *sgn3-3 myb36-2* shows the same pattern of suberisation as WT plants (Fig. 1e).

Our results indicate that MYB36 and SGN3 control endodermal lignification through two pathways: (a) a pathway involved in CS lignification controlled by both MYB36 and SGN3; and (b) a pathway involved in compensatory lignification of the endodermal cell corners controlled exclusively by SGN3.

**Cell-corner lignin is chemically distinct from CS lignin.** We investigated the chemical nature and biochemical origins of CS lignin and compensatory cell-corner lignin. For this, we used confocal Raman microscopy on root cross-sections, to spatially resolve the chemistry of these different types of lignin. We triggered endodermal cell-corner lignin deposition by feeding WT plants with CIF2 peptide (+CIF2), the ligand of the SGN3 receptor, in order to activate the Schengen-pathway. We separately imaged regions of interest containing CS lignin in WT

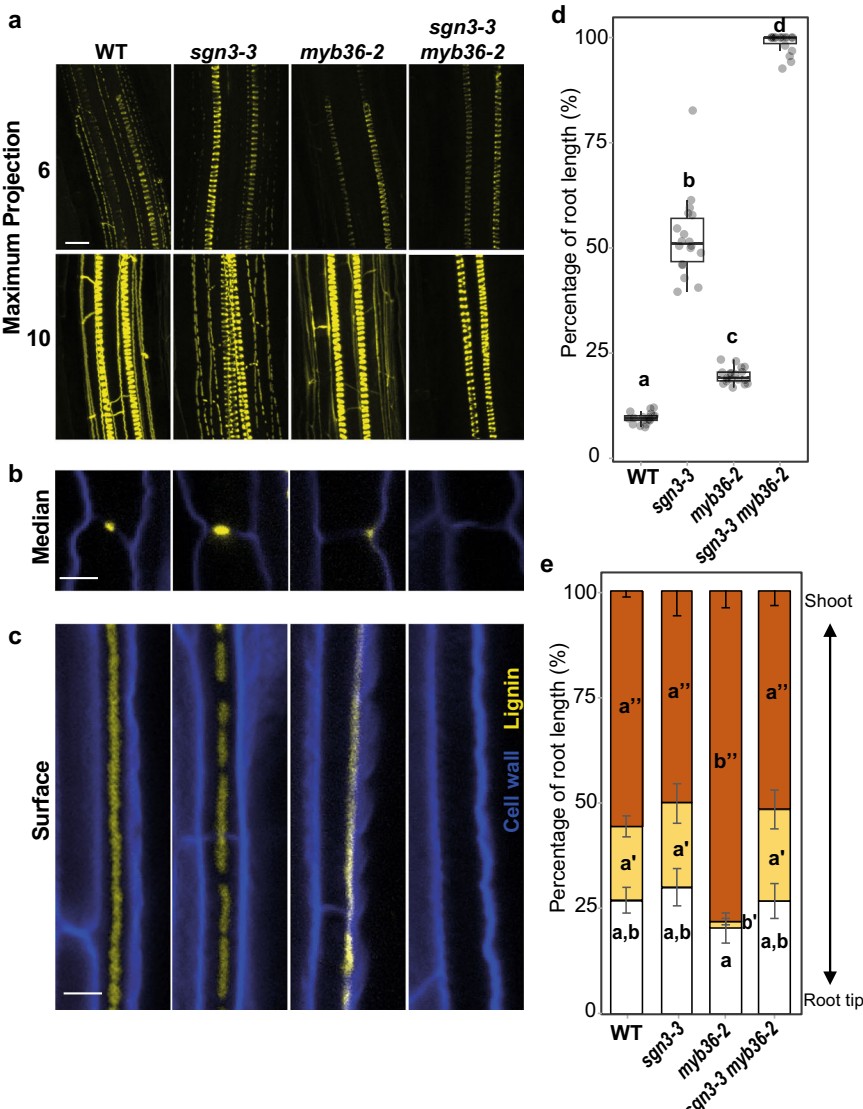

**Fig. 1 Disruption of *MYB36* and *SGN3* abolishes endodermal lignification and root apoplastic barrier. a** Maximum projection of lignin staining at the sixth and tenth endodermal cell after the onset of elongation. Spiral structures in the centre of the root are xylem. Scale bar = 10 μm. Median (**b**) and surface (**c**) view of an endodermal cell at ten cells after the onset of elongation. Scale bar = 5 μm. The roots were cleared and stained with basic fuchsin (yellow) for lignin and with Calcofluor white (blue) for cellulose. The experiment was repeated three times independently with similar results. **d** Boxplot showing the percentage of the root length permeable to propidium iodide. $n = 18$ from two independent experiments. Different letters represent significant differences between genotypes using a two-sided Mann–Whitney test ($p < 0.01$). Centre lines show the medians; box limits indicate the 25th and 75th percentiles. **e** Quantification of suberin staining along the root. The results are expressed in percentage of root length divided in three zones: unsuberized (white), discontinuously suberized (yellow), continuously suberized (orange). $n = 5$ plants for WT, $n = 6$ plants for *sgn3-3*, $n = 7$ plants for *myb36-2* and $n = 7$ plants for *sgn3-3 myb36-2*, error bars: SD, the centre of the error bars represents the mean. Individual letters show significant differences using a two-sided Mann–Whitney test between the same zones ($p < 0.01$). The experiment was repeated two times independently with similar results.

plants, endodermal cell-corner lignin in WT treated with CIF2, and xylem lignin from WT plants, treated or not with CIF2 (Supplementary Fig. 1). Then, we used multivariate curve resolution (MCR) analysis on these Raman images to spatially and spectrally resolve lignin in these different regions. We observed that the CS lignin spectrum is distinct from that of endodermal cell-corner lignin (Fig. 2a, b). For example, peaks known to be assigned to lignin display higher (ex: 1337 cm$^{-1}$, aliphatic OH bend[24]) and lower (ex: 1606 cm$^{-1}$, aromatic ring stretch[24]) intensity in CS lignin in comparison to endodermal cell-corner lignin. Another striking difference was observed for the peak at 1656–1659 cm$^{-1}$ assigned to a double bond conjugated to an aromatic ring (e.g.: coniferyl alcohol or coniferaldehyde,[24]). This

peak is missing in the endodermal cell-corner lignin of WT treated with CIF2 in comparison with CS lignin, suggesting a change in the phenolic composition of the cell-corner lignin. Conversely, the xylem lignin spectrum of plants treated with or without CIF2 was similar, with the most intense peaks showing comparable intensity. This suggests that changes in lignin composition triggered by the constitutive activation of the Schengen-pathway mainly occur in the endodermis, and xylem lignin remains largely unaffected.

These conclusions were further confirmed spatially by mapping the intensity of these different lignin spectra on large Raman maps containing xylem and endodermal lignin in WT plants treated or not treated with CIF2 (Fig. 2c). We observed that the

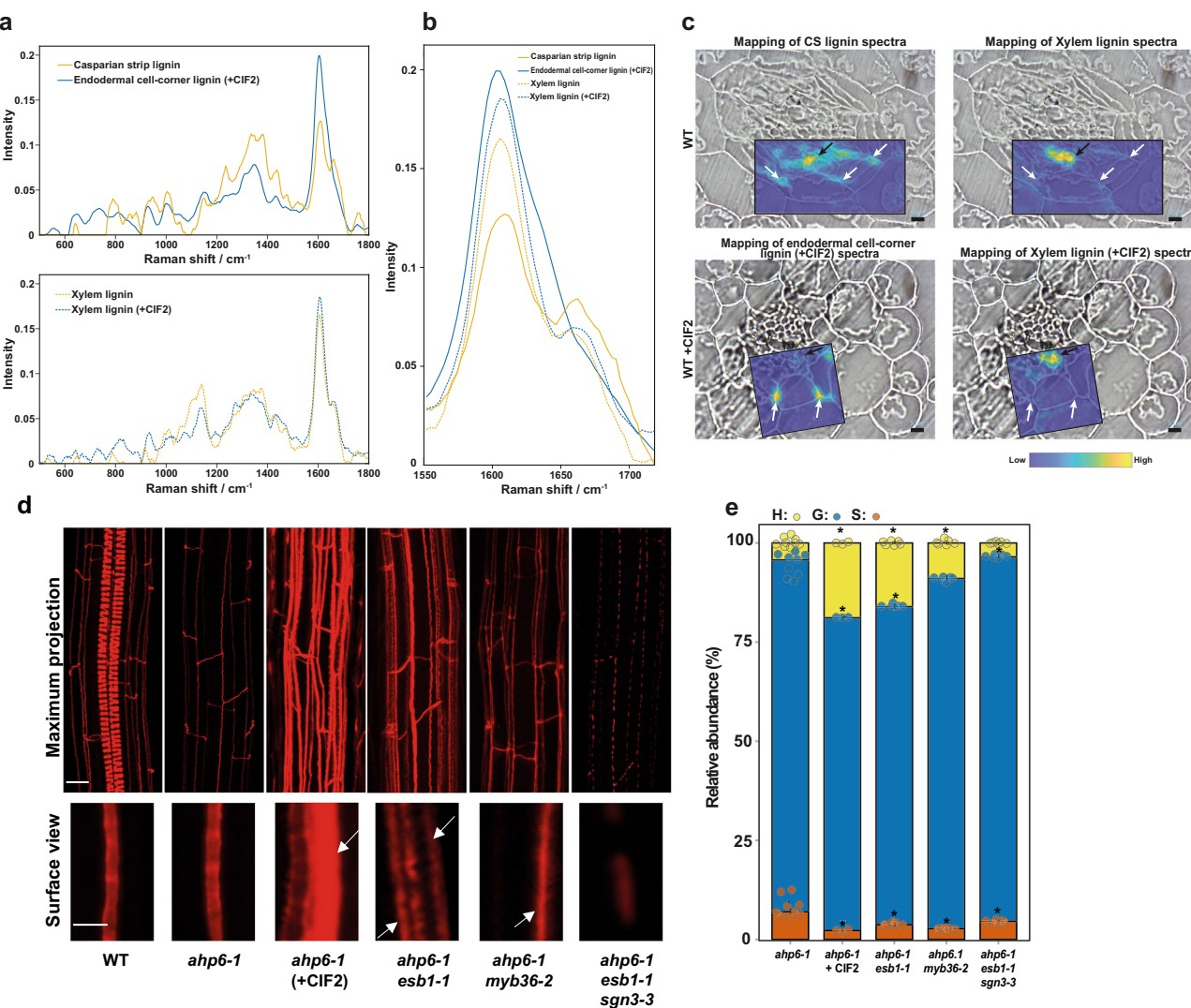

**Fig. 2 Activation of the Schengen-pathway triggers the deposition of a distinct "stress" lignin in the endodermis. a** Raman spectra of lignin of the different regions of interest presented in Supplementary Fig. 1 and determined using a Multivariate Curve Resolution (MCR) analysis. The MCR analysis was performed on small Raman maps from independent plants containing CS lignin of WT ($n = 8$), cell-corner lignin of WT treated with CIF2 (+CIF2; $n = 5$) and for xylem lignin of WT ($n = 2$) and xylem lignin of WT treated with CIF2 ($n = 2$). **b** Close view of Raman spectra presented in (**a**) in the lignin aromatic region between 1550 and 1700 $cm^{-1}$. **c** Large Raman maps in root cross-sections of WT and WT treated with CIF2 (+CIF2). The intensity of the different lignin spectra presented in (**a**) was mapped onto large Raman maps containing xylem and endodermal lignin. Scale bar = 5 μm. **d** Lignin staining with basic fuchsin at a distance of 3 mm from the root tip in WT, *ahp6-1*, *ahp6-1 esb1-1*, *ahp6-1 esb1-1 sgn3-3*, *ahp6-1 myb36-2* and *ahp6-1* treated with CIF2. The plants were grown for 6 days in presence of 10 nM 6-Benzylaminopurine (BA). Upper panel shows a maximum projection of the root (scale bar = 10 μm). Spiral structures in the centre only observed in the WT root are protoxylem. Lower panel shows surface view of endodermal cells. The experiment was repeated two times independently with similar results (scale bar = 5 μm). White arrows indicate ectopic lignification. **e** Relative abundance of the lignin monomers released by thioacidolysis (*p*-hydroxyphenyl (H), guaiacyl (G), and syringyl (S) units) in root tips of *ahp6-1* ($n = 9$), *ahp6-1* treated with CIF2 (+CIF2; $n = 3$), *ahp6-1 esb1-1* ($n = 6$), *ahp6-1 myb36-2* ($n = 6$) and *ahp6-1 esb1-1 sgn3-3* ($n = 6$). Asterisks represent significant differences from the *ahp6-1* control for each individual monomer using a two-sided Mann–Whitney *U* test ($p$ value < 0.05).

CS lignin spectrum localizes to the CS and xylem vessels suggesting a similar lignin composition, as previously shown for monomer composition using thioacidolysis[7]. Additionally, the endodermal cell-corner lignin spectrum localizes almost exclusively to the site of lignification in the corners of the endodermal cells and is essentially absent from the xylem. Furthermore, the xylem lignin spectrum in WT plants treated with CIF2 matches exclusively to the xylem vessel and is not observed at the endodermal cell corners. This strongly supports the conclusion that constitutive activation of the Schengen-pathway triggers the deposition of lignin at endodermal cell corners that has a different chemical composition compared to both CS and xylem lignin.

To confirm these differences between CS and endodermal cell-corner lignin, we adopted an approach to directly measure the subunit composition of endodermal lignin avoiding possible contamination from the highly lignified protoxylem cells[7]. We genetically crossed a collection of CS mutants that represent a different level of lignin accumulation in the endodermis with the *arabidopsis histidine transfer protein 6.1* mutant (*ahp6-1*). This mutant, in the presence of low amounts of the phytohormone cytokinin, shows a strong delay in protoxylem differentiation, without affecting CS formation (Fig. 2d)[7,25]. Therefore, in the resulting lines the majority of lignin derived from the protoxylem is not present allowing us to analyze primarily

lignin with an endodermal origin. To explore how the chemical composition of the cell-corner lignin differs from CS lignin, we collected root tips (3 mm) of 6-day-old *ahp6-1* and *ahp6-1 esb1-1 sgn3-3* mutants accumulating CS lignin only, and from mutants (*ahp6-1 myb36-2* and *ahp6-1 esb1-1*) with cell-corner lignification and a reduced amount of CS lignin. Additionally, as a control we used *ahp6-1* plants treated with the CIF2 peptide that strongly induces the Schengen-pathway and the deposition of cell-corner lignin (Fig. 2d). We measured the relative content of H, G and S subunits in lignin from all samples using thioacidolysis followed by GC-MS (Fig. 2e). We found that CS lignin monomer composition in our control line *ahp6-1* (H: 5%, G: 87%, S: 8%) was similar to that previously reported[7]. The monomer composition of the defective CS in the mutant *ahp6-1 esb1-1 sgn3-3* is overall similar to WT with a small increase in G and decrease in S subunits. Strikingly, we observed that lignin composition in the lines and treatments that induce the accumulation of cell-corner lignin (*ahp6-1 esb1-1, ahp6-1 myb36-2, ahp6-1*(+CIF2)) was different from the control and mutant lines that only accumulate lignin in the CS. Lignin extracted from plants accumulating cell-corner lignin showed a higher proportion of H subunits. In the case of *ahp6-1* treated with CIF2, H content was increased to 19% and G content was decreased to 79%.

Such a high content of H subunits in lignin is rarely found in angiosperm. Similar levels of H subunits in lignin mainly occur in compression wood of gymnosperm[26–29] and in defence-induced lignin and have been termed "stress lignin"[26,30–33]. We therefore conclude that Schengen-pathway induced endodermal cell-corner lignin is a form of "stress lignin". Taken together, both chemical analysis of lignin subunits by thioacidolysis and spatially resolved confocal Raman spectroscopy show that lignin deposited in endodermal cell corners upon constitutive activation of the Schengen-pathway is H-rich, and chemically and spatially distinct from both CS and xylem lignin.

**Cell-corner lignin composition controls root permeability**. We then try to determine if the compositional change of cell-corner lignin has functional consequences. For this, we used *myb36-2* displaying cell-corner lignin only and no CS lignin (Fig. 1b, c). Using a pharmacological approach (Supplementary Fig. 2a), we blocked endogenous monolignol production with an inhibitor of the phenylpropanoid pathway, piperonylic acid (PA)[7]. The PA treatment led to a strong reduction of cell-corner lignin deposition (Supplementary Fig. 2b, c) and a strong increase of root permeability as determined with the apoplastic tracer PI (Supplementary Fig. 2d). We then attempted to chemically complement the PA-induced defects by exogenous application of each of the three canonical monolignols *p*-coumaryl (for H subunit), coniferyl (for G subunit), and sinapyl (for S subunit) alcohols and by combining the two main monolignols *p*-coumaryl and coniferyl alcohols that are incorporated into cell-corner lignin (Fig. 2e). The exogenous application of *p*-coumaryl or coniferyl alcohols can slightly trigger cell-corner lignification but, sinapyl alcohol did not (Supplementary Fig. 2b, c). The combined application of *p*-coumaryl and coniferyl alcohols increased the deposition of cell-corner lignin in comparison with their individual application. None of the three monolignols alone can fully recover the PA-induced defect on root permeability (Supplementary Fig. 2d). The combined application of *p*-coumaryl and coniferyl alcohols fully recovered the effect of PA on root permeability in the *myb36* mutant. This chemical complementation assay indicates that the type of monolignols available controls both cell-corner lignin deposition, and its capacity to seal the endodermal apoplast. The combination of *p*-coumaryl and

coniferyl alcohols is the most effective for increasing lignin deposition and for sealing the apoplast.

**Schengen control of the phenylpropanoid pathway**. To investigate the biosynthesis of the endodermal H-rich stress lignin, we performed RNA-seq using root tips (5 mm) of WT plants, WT treated with exogenous CIF2, and the mutants *myb36-2* and *esb1-1* that show activation of the Schengen-pathway, and root tips of *sgn3-3*, *esb1-1 sgn3-3*, *sgn3-3 myb36-2* and *sgn3-3* treated with exogenous CIF2 with no Schengen signalling. Clustering analysis of the differentially expressed genes (DEGs) shows that roots displaying cell-corner lignification due to the constitutive activation of the Schengen-pathway (WT treated with exogenous CIF2, *myb36-2* and *esb1-1*) share a similar transcriptional response that is distinct from that observed in the other genotypes (Fig. 3a, Supplementary Fig. 3a and Supplementary Data 1). CIF2 application to *sgn3-3* shows a similar transcriptional response to non-treated WT and *sgn3-3* and does not trigger the transcriptional changes observed during the activation of the Schengen-pathway (WT treated with exogenous CIF2, *myb36-2* and *esb1-1*). This is in line with previous published transcriptomic analysis using plant treated with CIF2[18], and the idea that SGN3 is the only receptor for CIF2 in roots. We observed that genes in cluster C1 are upregulated by the activation of the Schengen-pathway. This cluster is enriched in genes involved in the phenylpropanoid pathway (Supplementary Fig. 3b) and contains a large set of peroxidases and laccases (Supplementary Fig. 3c). We hypothesized that the activation of this pathway would provide the phenolic substrates that are subsequently polymerized by laccases and peroxidases for the enhanced lignification and suberisation induced by the Schengen-pathway. We observed strong activation of expression of genes encoding all the key enzymes of the phenylpropanoid pathway required for monolignol biosynthesis, with the exception of *C3′H, C3H, HCT* and *F5H* (Fig. 3b and Supplementary Fig. 3d). H-rich lignin is known to be accumulated when expression of *C3′H* or *HCT* is repressed in *A. thaliana* and poplar[35–40]. This activation of all the main enzymes of the phenylpropanoid pathway, apart from *C3′H* and *HCT*, observed after triggering the Schengen-pathway, could explain the high level of H-units incorporation into endodermal cell-corner lignin (Fig. 3b and Supplementary Fig. 3d). Similarly, the roots of the cellulose synthase isomer mutant *ectopic lignification1* (*eli1*) accumulate H-rich lignin and display strong gene activation for most of the phenylpropanoid pathway, with the exception of *C3′H*[34]. Interestingly, ectopic lignification in *eli1* is also under the control of another receptor-like kinase, *THE1* (THESEUS), also involved in CW integrity sensing[35,36].

We then tried to identify transcriptional regulators with a role in the Schengen-pathway controlled regulation of phenylpropanoid synthesis. We performed a gene expression correlation analysis between the phenylpropanoid pathway genes and their transcriptional regulators[3,37] (Supplementary Fig. 3d). We found that the expression of the transcription factor *MYB15* highly correlates with the expression of most of the genes required for monolignol biosynthesis, with the notable exception of *C3′H* (Supplementary Fig. 3d). Upregulation of *MYB15* in response to CIF2 has been previously shown[18]. This transcription factor is known to bind to the promoter of *PAL1, C4H, HCT, CCoAOMT1* and *COMT* but does not bind to the promoter of *C3′H* and *F5H*[38]. Schengen-pathway activation of *MYB15* expression provides a plausible mechanism to explain the induction of the main enzymes of the phenylpropanoid pathway with the exception of *C3′H* and *F5H*. This modulation of gene expression could explain the enhanced incorporation of *p*-coumaryl alcohol into the stress lignin we observed at

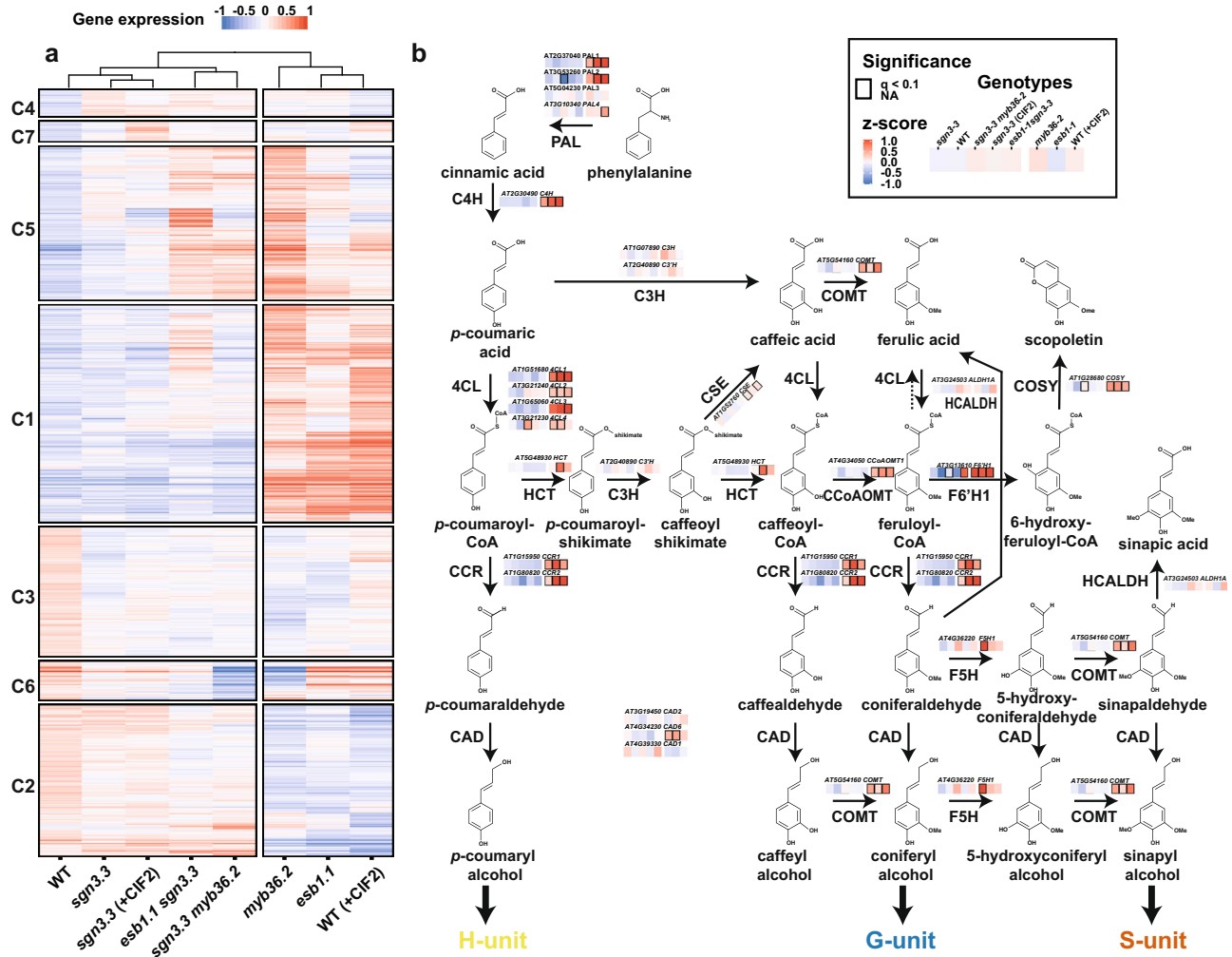

**Fig. 3 Modulation of the phenylpropanoid pathway by the Schengen-pathway. a** Heatmap of the 3564 differentially expressed genes identified in the RNA-seq in root tips of wild-type (WT), *sgn3-3*, *esb1-1*, *myb36-2*, *esb1-1 sgn3-3*, *sgn3-3 myb36-2* plants. Treatment with 100 nM CIF2 was applied as indicated (+CIF2) for WT and *sgn3-3* plants. Clusters (C) are designated with numbers (n = 7). Genes belonging to each cluster are listed in Supplementary Data 1. **b** Phenylpropanoid pathway leading to the lignin monomers and scopoletin biosynthesis (adapted from[72]). Solid arrows represent enzymatic steps. Gene expression from the genes selected in Supplementary Fig. 3d was mapped on the pathway according to their KEGG enzyme nomenclature. Only the genes with a demonstrated function in lignin biosynthesis as listed in Supplementary Fig. 3d were mapped. PAL PHENYLALANINE AMMONIA-LYASE, C4H CINNAMATE 4-HYDROXYLASE, 4CL 4-COUMARATE:CoA LIGASE, HCT *p*-HYDROXYCINNAMOYL-CoA:QUINATE/SHIKIMATE *p*-HYDROXYCINNAMOYLTRANSFERASE, C'3H *p*-COUMARATE 3'-HYDROXYLASE, C3H COUMARATE 3-HYDROXYLASE, CSE CAFFEOYL SHIKIMATE ESTERASE, CCoAOMT CAFFEOYL-CoA O-METHYLTRANSFERASE, CCR CINNAMOYL-CoA REDUCTASE, F5H FERULATE 5-HYDROXYLASE, COMT CAFFEIC ACID O-METHYLTRANSFERASE, CAD CINNAMYL ALCOHOL DEHYDROGENASE, HCALDH HYDROXYCINNAMALDEHYDE DEHYDROGENASE, COSY COUMARIN SYNTHASE, F6'H1 FERULOYL COA ORTHO-HYDROXYLASE 1.

endodermal cell corners. Interestingly, increased incorporation of *p*-coumaryl alcohol into lignin has been found in response to *Pseudomonas syringae* and is partially controlled by the MYB15 transcription factor[39]. MYB15 is an activator of basal immunity in *A. thaliana* through induction of the synthesis of defence lignin and soluble phenolics[38].

To test if constitutive activation of the Schengen-pathway leads to the production of defence-inducible soluble phenolics, we undertook secondary metabolite profiling using ultra high performance liquid chromatography (UHPLC). This analysis was performed using root tips (5 mm) of the *esb1-1* mutant having a defective CS and constitutive activation of the Schengen-pathway, *sgn3-3* and *sgn3-3 esb1-1* mutants having a defective CS and inactivation of the Schengen-pathway and in WT plants. We observed distinct accumulation of soluble secondary metabolites across the different genotypes (Supplementary Fig. 4 and Supplementary Data 2). We

identified 20 out of 52 phenolic compounds that differentially accumulate specifically due to the activation of the Schengen-pathway. We found higher accumulation of the conjugated neolignan G(8-O-4)*p*CA, scopoletin, flavonoid derivatives such as conjugated kaempferol (astragalin and 4'-O-acetylkaemferol-3-O-hexoside), isorhamnetin and acetylhyperoside. Scopoletin biosynthesis is controlled by the enzyme F6'H1 and COSY[40,41] and the transcription factor MYB15[38]. We found that the expression of the three genes encoding these proteins is induced by the constitutive activation of the Schengen-pathway (Fig. 3b and Supplementary Fig. 3d). Scopoletin is a modulator of plant-microbe interaction[45,49–52]. In addition to that, we found a strong induction of genes related to defence (response to chitin/systemic acquired resistance/immune response/hypersensitive response) among the genes induced by the activation of the Schengen-pathway (C1; Fig. 3a and Supplementary Fig. 3b). This is consistent with a previous publication showing

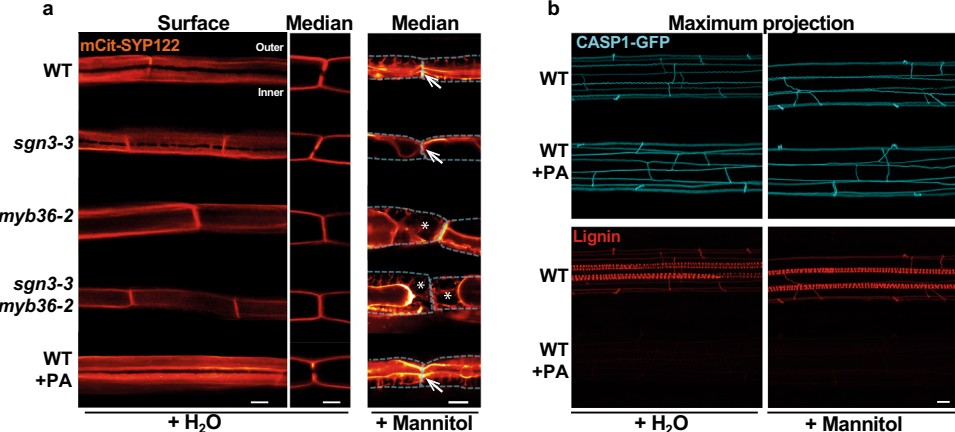

**Fig. 4 PM attachment to the CW is MYB36-dependent but does not rely on lignin deposition. a** Median and surface view of the endodermal plasma membrane using the marker line pELTP::SYP122-mCitrine before plasmolysis (+H₂O) and after plasmolysis (+Mannitol) at 15 cells after the onset of elongation. WT plants were treated or not from germination with 10 μM piperonylic acid (+PA). White asterisks show the exclusion domain at the CSD. The dashed line represents the contours of the cells before plasmolysis. Arrows show the plasma membrane attachment to the cell wall. Blue asterisks show the plasmolysis generated space where no attachment is observed. Scale bar = 5 μm. "inner" designates the stele-facing endodermal surface, "outer", the cortex-facing surface. The experiment was repeated three times independently with similar results. **b** Maximum projection of CASP1-GFP and lignin staining with basic fuchsin in cleared roots from plants grown with or without 10 μM piperonylic acid and subjected to plasmolysis with Mannitol. Scale bar = 10 μm. The experiment was repeated two times independently with similar results.

similarities between the Schengen-pathway and the microbe-associated molecular patterns signalling pathway[18].

**Local activation of genes related to defence**. We then tried to determine if this induction of genes related to defence occurs locally or systemically. For this, we used a split-root system, in which the roots of a single plant were physically separated for 3 days. One half of the root system was exposed to the Schengen-pathway activator CIF2, whereas the other side was kept in the same medium without CIF2 (Supplementary Fig. 5a). We observed that cell-corner lignification is deposited only in the root with direct contact with CIF2 (Supplementary Fig. 5b) as observed for the induction of the expression of peroxidase genes (Supplementary Fig. 5c). Similarly, several genes related to defence were found to be induced only locally in the presence of CIF2 (Supplementary Fig. 5d).

The constitutive activation of the Schengen-pathway and subsequent cell-corner lignification described here are triggered locally when the integrity of the endodermal apoplastic barrier is lost. This can also occur during developmental processes such as lateral root emergence and during infection with pathogens. Interestingly, the deposition of endodermal barriers has been associated with increased resistance against a large range of soil-borne pathogens such as *Aphanomyces euteiches*[42], *Ralstonia solanacearum*[43], *Phytophthora sojae*[44] and nematodes[45]. The resistance to *A. euteiches*, *R. solanacearum* is also accompanied by the production of soluble phenolics[42,43].

**Cell wall attachment to plasma membrane (PM) relies on CS domain**. The apoplast in between two endodermal cells is sealed by the deposition of CS lignin. This sealing is perfected by the anchoring of the CS membrane domain (CSD) to the CW, through an unknown mechanism. Upon plasmolysis, the protoplasts of endodermal cells retract but the CSD remains tightly attached to the CS[23,46,47]. This attachment appears in a developmental manner during the differentiation of the endodermis. It occurs concomitantly with the recruitment of the CASPs at the CSD, and with CS lignin deposition[23]. We then wanted to study whether or not the different types and sites of lignification contribute to the attachment of the PM, to the CW. To visualize the

PM, we introduced an endodermis-specific PM marker (pELTP::mCit-SYP122) in WT, *sgn3-3*, *myb36-2* and *sgn3-3 myb36-2* backgrounds (Fig. 4a). The PM marker is excluded from the CSD in WT as described for other endodermal PM marker lines[13,23]. This exclusion is still observed in *sgn3-3* but in an interrupted manner similarly to that observed for lignin (Fig. 1a–c). The exclusion domain disappears entirely in *myb36-2* and *sgn3-3 myb36-2*. Additionally, no exclusion zone in the PM is observed in *myb36-2* where cell-corner lignin is deposited.

We then used mannitol-induced plasmolysis to visualize the PM attachment to the CW. Upon plasmolysis, the PM retracts but remains attached to the CS in WT and *sgn3-3*, forming a flattened protoplast (Fig. 4a). However, small portions of the PM are able to detach from the CW in a *sgn3-3* mutant as seen in Supplementary Fig. 6. This is likely to happen where the PM exclusion domain is interrupted in *sgn3-3* (Fig. 4a). In *myb36-2* and *sgn3-3 myb36-2*, the CW attachment to the PM is lost (Fig. 4a and Supplementary Fig. 6). Importantly, retraction of the PM is observed in *myb36-2* at the corner of the endodermal cells on the cortex side where cell-corner lignin is deposited (Fig. 1a). These results establish the requirement of MYB36 for the formation of the CSD excluding the PM marker. Additionally, the presence of CSD, but not cell-corner lignin, is required for PM attachment to the CW.

We then tested if CS lignin is required for the PM attachment to the CS. For this, we used an inhibitor of the phenylpropanoid pathway, PA. Treatment with PA suppresses lignin accumulation in the vasculature and in the CS (Fig. 4b). Absence of lignin did not affect the exclusion of the PM marker at the CSD. This was further confirmed using the CSD marker line pCASP1::CASP1-GFP[12] that showed similar localization independently of the CS lignin presence (Fig. 4b). Additionally, the PM attachment to the CS is still observed when CS lignin deposition is inhibited (Fig. 4a, b and Supplementary Fig. 6). These findings confirm previous reports[13,18] showing that CS lignin is not required for the formation of the CSD. Importantly, these results indicate that CS lignin does not participate in anchoring the CSD to the CW. Other CW compounds might be involved in that process.

The absence of PM attachment to the site of cell-corner lignification is likely to affect the permeability of the apoplast of

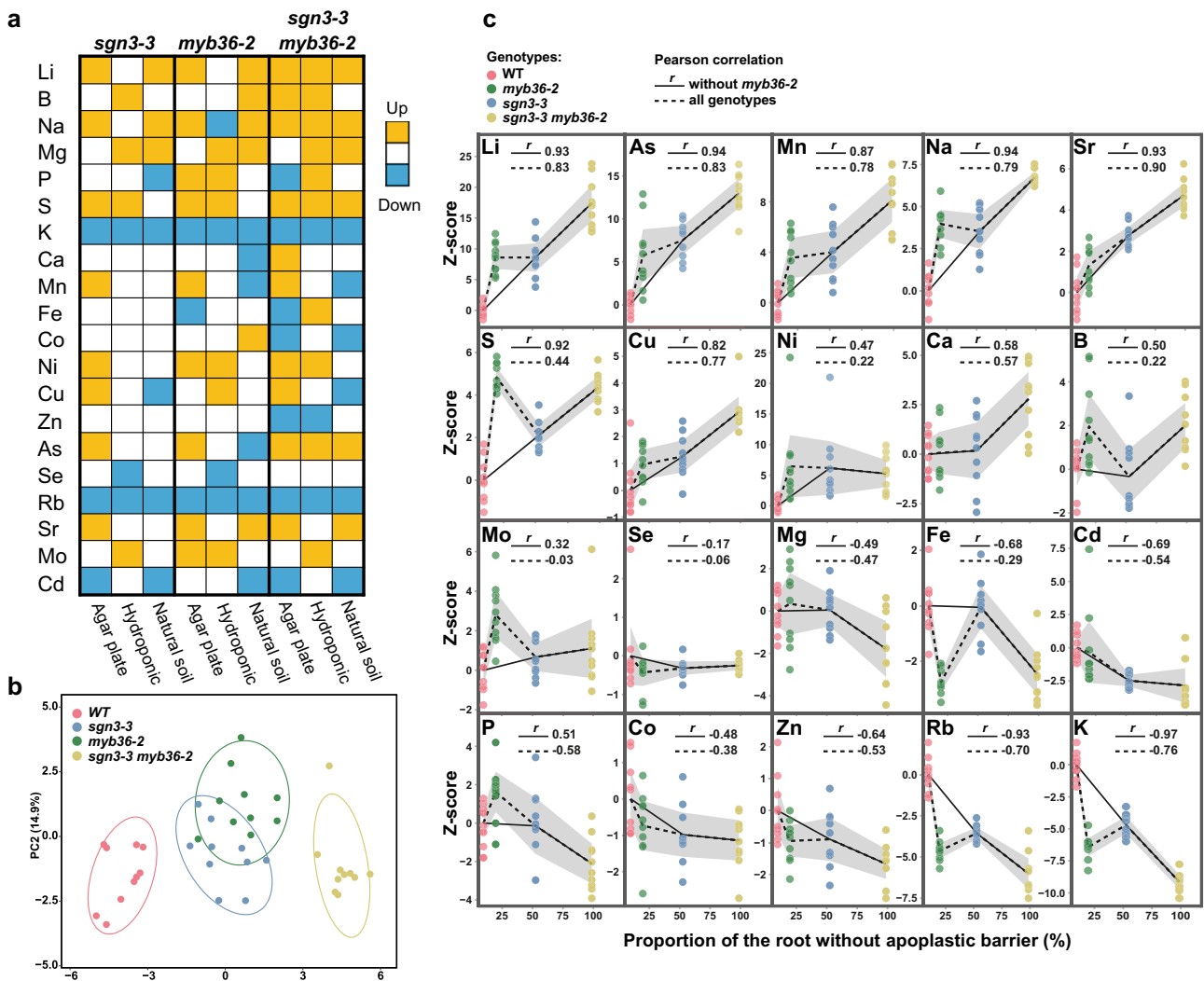

**Fig. 5 Absence of endodermal apoplastic barrier triggers major ionomic changes. a** Overview of ions accumulation in shoot of *sgn3-3*, *myb36-2* and *sgn3-3 myb36-2* mutants compared to WT using different growth conditions in agar plates (long day, *n* = 10 individual plants), in hydroponics (short day, *n* = 6 individual plants) and natural soil (short day, *n* = 18 for WT, *n* = 18 for *sgn3-3*, *n* = 18 for *myb36-2* and *n* = 13 for *sgn3-3 myb36-2*). Element concentration was determined by ICP-MS and is available in Supplementary Data 3. Colour code indicates significant changes in accumulation compared with the WT using a two-sided *t*-test (*p* < 0.01). **b** Principal component analysis (PCA) based on the concentration of 20 elements in shoots of plants grown in agar plates. Ellipses show confidence level at a rate of 90%. *n* = 10 individual plants. **c** Plots presenting the correlation between the *z*-scores of elements content in shoots of plants grown in agar plates of WT, *myb36-2*, *sgn3-3* and *sgn3-3 myb36-2* against the portion of root length permeable to propidium iodide as determined in Fig. 1d. The black lines show the average and the grey area show the 95% confidence interval (*n* = 10 individual plants).

the endodermal cells. This can consequently affect the transport of water and solutes to the shoot.

**Absence of apoplastic barrier triggers major ionomic changes.** The mutants described in our previous analyses display differential patterns of lignin deposition and composition, and this consequently affects root apoplastic permeability. This affords a unique opportunity to assess the role of endodermal lignification in controlling nutrient homoeostasis in the plant. The *sgn3-3* (delayed CS barrier, no cell-corner lignin), *myb36-2* (no CS lignin, has cell-corner lignin) and *sgn3-3 myb36-2* (no CS or cell-corner lignin) mutants were grown using different growth conditions (agar plate, hydroponic and natural soil) and their leaves were analyzed for their elemental composition (ionome) using inductively coupled plasma-mass spectrometry (ICP-MS) (Fig. 5a and Supplementary Data 3). A principal component analysis (PCA) of the ionome of leaves reveals that all the mutants have different leaf ionomes compared to WT when grown on plates (Fig. 5b),

hydroponically and to a lesser extent in a natural soil (Supplementary Fig. 7a, b). Based on the PC1 axis, *sgn3-3 myb36-2* displayed the most distinct ionomic phenotype (Fig. 5b and Supplementary Fig 7a, b). In line with our previous results (Fig. 1d), this effect indicates an additivity of the two mutations on the leaf ionome. Importantly, this result also supports the idea that cell-corner lignin in *myb36* can act as an apoplastic barrier to mineral nutrients.

We next tested the correlation between the gradient of root apoplastic permeability across WT, *myb36-2*, *sgn3-3* and *sgn3-3 myb36-2* determined in Fig. 1d, with their leaf elemental content (Fig. 5c). We observed that *myb36-2* does not fit into this correlation analysis as well as the other genotypes. This is likely due to activation of the Schengen-pathway leading to deposition of endodermal cell-corner stress lignin, early suberisation, reduced root hydraulic conductivity, activation of ABA signalling in the shoot and stomata closure known to occur in this mutant[9,48]. Additionally, the *myb36-2* mutation interferes with

overall root development (Supplementary Fig. 7c–e) as previously reported[49]. This is due to the constitutive activation of the Schengen-pathway as *sgn3-3 myb36-2* shows normal root development. Removal of *myb36-2* from the correlation analysis, leaving just lines with an inactive Schengen-pathway, improved the Pearson correlation coefficient for almost all the elements, and we observed a strong correlation ($r \geq 0.5$ or $\leq -0.5$) for 15 out of the 20 elements. We observed a strong positive correlation between an increased CS permeability and leaf accumulation of lithium (Li), arsenic (As), manganese (Mn), sodium (Na), strontium (Sr), sulfur (S), copper (Cu), calcium (Ca), boron (B) and a strong negative correlation with iron (Fe), cadmium (Cd), phosphorus (P), zinc (Zn), rubidium (Rb) and potassium (K). This suggests that a functional apoplastic barrier is required to limit the loss of essential elements such as K, Zn, Fe and P. Conversely, a defective apoplastic barrier allows increased leaf accumulation of the essential nutrients Mn, S, Cu, Ca and B. These gradients of higher and lower accumulations of mineral nutrients and trace elements illustrate the bidirectional nature of the CS barrier, by blocking some solutes from entering the vasculature, and by facilitating the accumulation of other solutes in the stele for translocation.

**Casparian strips do not control root hydraulic conductivity.** We then measured the capacity of the root to transport water, also called root hydraulic conductivity ($Lp_r$), in 3-week-old plants grown hydroponically. We observed that the root hydraulic conductivity remains unchanged in *sgn3-3* and *sgn3-3 myb36-2* in comparison with WT (Fig. 6a). In contrast, *myb36-2* showed a strong reduction of root hydraulic conductivity. These results established that the CS-based endodermal apoplastic seal does not control root water transport capacity, as in the absence of any barriers in *sgn3-3 myb36-2* (Fig. 1d) root hydraulic conductivity is the same as WT. This is consistent with water transport occurring mainly via the transcellular pathway, with a major contribution via aquaporins[50]. The reduced hydraulic conductivity observed in *myb36-2* is consistent with that previously observed in *esb1*, which also has an activated Schengen-pathway[48]. The reduced hydraulic conductivity in *esb1* originates mainly from a reduction in aquaporin-mediated water transport as determined using a pharmacological approach[48]. Here, our RNA-seq experiment revealed a GO-term enrichment in cluster C2 (genes repressed by the Schengen-pathway, Fig. 3a) relating to water deprivation (Supplementary Fig. 3b) and importantly, ten aquaporin genes are down regulated by activation of the Schengen-pathway (Fig. 6b). This set of aquaporin genes contains several highly expressed aquaporins in root, including *PIP2,2* known to significantly contribute to root hydraulic conductivity[51,52]. This provides an explanation for the reduction in root hydraulic conductivity observed in both *myb36* and *esb1*.

**Endodermal lignification and plant fitness.** Given the significant impacts that CS and Schengen-pathway activation have on mineral nutrient homoeostasis (Fig. 5) and water transport (Fig. 6a, b), we further investigated their impact on growth and development. The double mutant *sgn3-3 myb36-2* displayed a severe dwarf phenotype when grown hydroponically or in natural soil but not on agar plates, in comparison with WT and the single mutants (Supplementary Fig. 7c–g). This indicates a critical role of CS for maintaining normal plant growth and development. However, this is conditioned by the growth environment. The high humidity environment and consequently reduced transpiration of plants on agar plates in comparison with the other growth environments could explain these phenotypical differences. Indeed, reduced leaf transpiration is a key part of the

compensation mechanisms mitigating the loss of CS integrity, allowing relatively normal growth as previously reported[48]. We then tested if differences in relative humidity (RH) can affect plant growth in the absence of an endodermal root barrier. For this, we used *sgn3-3* (delayed CS barrier, no cell-corner lignin), *myb36-2* (no CS lignin, has cell-corner lignin) and *sgn3-3 myb36-2* (no CS and no cell-corner lignin). Additionally, we tested if the presence of endodermal suberin can affect plant growth by using lines expressing the Cutinase DEstruction Factor (CDEF) under the control of an endodermis-specific promoter (pELTP::CDEF) in a WT and *sgn3-3 myb36-2* background. Lines expressing CDEF show a strong reduction in endodermal suberin deposition (Supplementary Fig. 8a). Seedlings were germinated and grown in soil in a high humidity environment (80% RH) for 7 days, and then transferred to an environment with the same (80% RH) or lower (60% RH) humidity. We measured the leaf surface area as a proxy of plant growth[53] at 0, 2, 5 and 8 days after transfer (Fig. 6c and Supplementary Fig. 8b). In both the high and low humidity environment, all mutants with reduced CS functionality (*sgn3-3*, *myb36-2*, *sgn3-3 myb36-2* and *sgn3-3 myb36-2-pELTP::CDEF*) show a reduction of leaf surface in comparison with WT. Importantly, the growth reduction observed in the absence of endodermal lignification (*sgn3-3 myb36-2*) is severe, specifically in low humidity conditions, in comparison with all other genotypes and high humidity conditions. The *sgn3-3 myb36-2* plants with no growth after 9 days started to display necrosis over all the leaf surface and were considered dead as quantified in Fig. 6d. Low humidity triggers a high percentage of mortality in *sgn3-3 myb36-2* and to a lesser extent in *sgn3-3* compared to WT and to the other genotypes in which no mortality is observed when grown in low humidity conditions. Such mortality was not observed when *sgn3-3 myb36-2* was grown in high humidity. This highlights that endodermal lignification is required for maintaining plant growth and survival under low humidity. However, this is not the case for endodermal suberisation, because the removal of suberin through expression of *CDEF* in WT and *sgn3-3 myb36-2* did not affect mortality or leaf surface area after 8 days at a lower humidity in comparison with their respective backgrounds (Fig. 6c, d and Supplementary Fig. 8b).

The strong growth reduction observed in *sgn3-3 myb36-2* in comparison with WT and the single mutants could be associated with a lack of root selectivity leading to major ionomic changes as shown in Fig. 5. Low humidity would generate a higher transpiration stream and consequently more uncontrolled, and potentially detrimental, accumulation of mineral nutrient and trace elements in the leaves, compared with high humidity. Conversely, high humidity would slow transpiration rate, allowing plants to better control nutrient acquisition[48]. To test this, we measured elemental accumulation in leaves of WT, *sgn3-3*, *myb36-2* and *sgn3-3 myb36-2* exposed to 80% RH or 60% RH for 5 days (Supplementary Fig. 8c and Supplementary Data 3). In WT, we observed that 60% RH triggers a decrease in the concentration of Cd, Zn, K, (and chemical analogue Rb), and Ca (and chemical analogue Sr) with a similar trend observed in *myb36.2* for these elements. In contrast, in *sgn3-3 myb36-2*, lower humidity did not trigger such decreases but rather caused an increased accumulation for K (and Rb) and Na (and the chemical analogue Li). This shows that endodermal lignification is essential for the plants to adjust their nutrient balance in response to low humidity. The uncontrolled accumulation of Na could contribute to the growth defect observed in *sgn3.3 myb36.2*.

We then measured the impact of the absence of CS, suberin or of the constitutive activation of the Schengen-pathway on plant fitness. For this, we determined the number of seed-containing siliques per plant as an estimation of fitness (Fig. 6e). A

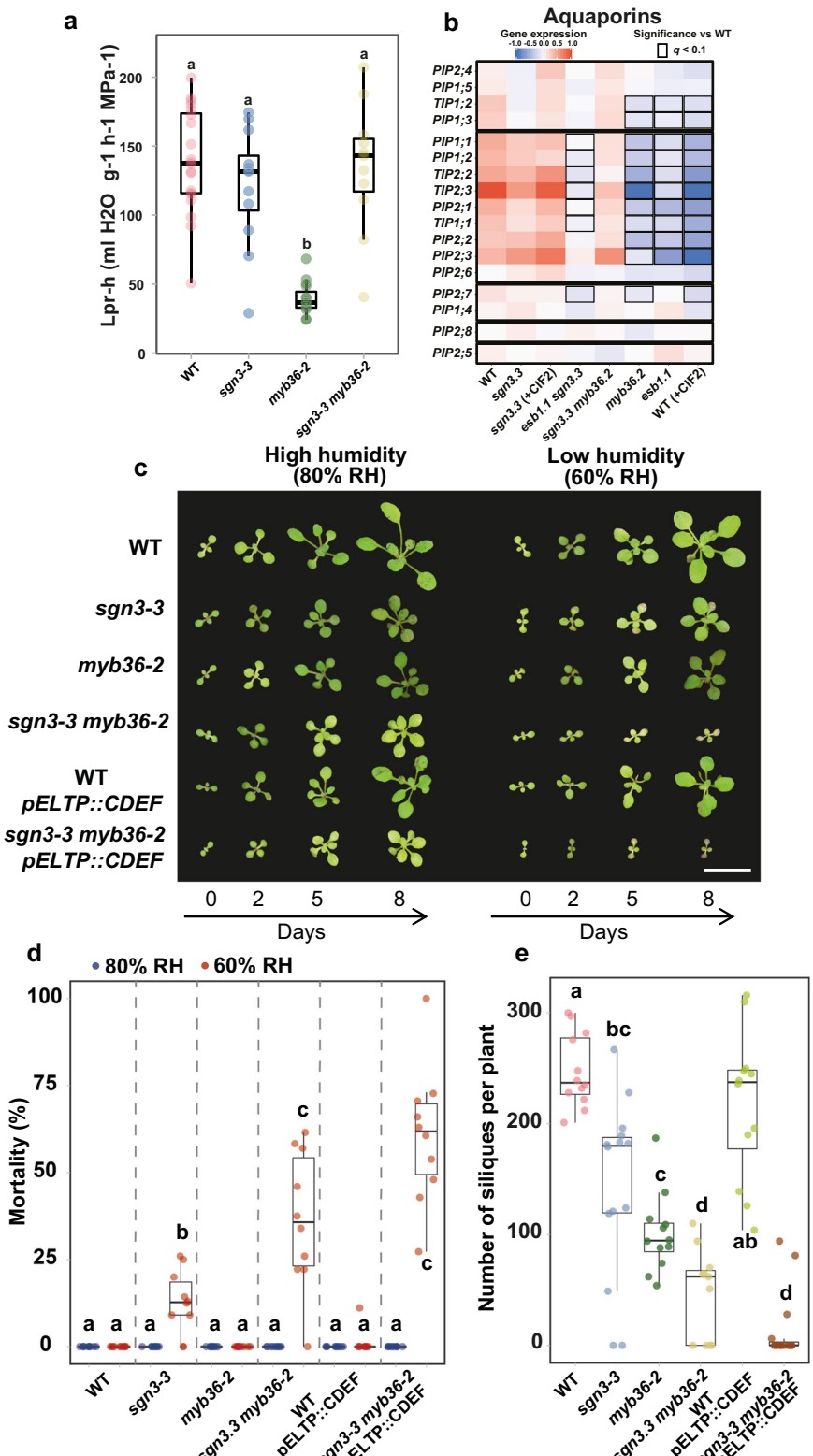

significant reduction of silique numbers is observed in all the genotypes in comparison to WT, with the exception of *pELTP:: CDEF* in the WT background. The *sgn3-3* mutant, with a partial root apoplastic barrier defects, showed a decrease in siliques number in comparison with WT. A similar decrease was observed for *myb36-2* displaying also a partial root apoplastic barrier defect and with cell-corner stress-lignin deposition. Complete disruption of endodermal lignification strongly affects

silique production as observed for *sgn3-3 myb36-2* and *sgn3-3 myb36-2—pELTP::CDEF*. These results clearly establish that the CS is essential for plant fitness. Furthermore, activation of the Schengen-pathway helps protect the plant from the detrimental impact on fitness when the barrier function of the CS is compromised. The *sgn3-3 myb36-2—pELTP::CDEF* line reported here, with it complete lack of endodermal lignin and suberin extracellular barriers, and Schengen-dependent signalling, is a

**Fig. 6 Activation of the Schengen-pathway represses water transport and maintains plant growth, survival, and fitness under fluctuating environment.**
**a** Boxplot showing the hydrostatic root hydraulic conductivity ($Lp_{r-h}$) in WT, sgn3-3, myb36-2, sgn3-3 myb36-2 grown hydroponically for 19–21 days under environmental controlled conditions. Hydraulic conductivity was measured using pressure chambers ($Lp_{r-h}$). Different letters indicate significant differences between genotypes determined by an ANOVA and Tukey's test as post hoc analyses ($p < 0.01$, $n = 20$ for WT, $n = 12$ for sgn3-3, $n = 15$ for myb36-2 and $n = 11$ for sgn3-3 myb36-2). Centre lines show the medians; box limits indicate the 25th and 75th percentiles. The experiment was repeated two times independently with similar results. **b** Heatmap of aquaporins expression across the different genotypes and treatments used in the RNA-seq experiment. **c** Representative pictures of WT, sgn3-3, myb36-2, sgn3-3 myb36-2, WT—pELTP::CDEF and sgn3-3 myb36-2—pELTP::CDEF plants germinated in soil with a high humidity (80%) for 7 days and then transferred in an environment with a lower (60% RH) or with constant humidity (80% RH). Pictures were taken 0, 2, 5 and 8 days after the transfer. Scale bar = 1 cm. The experiment was repeated two times independently with similar results. **d** Boxplots showing the proportion of dead plants after transfer in an environment with constant humidity (80% RH, blue) or with a lower (60% RH, red). The plants displaying no growth after 9 days and showing necrosis in all the leave surface were considered as dead plants. Each point represents the proportion of dead plants in a cultivated pot compared to the total number of plants for one genotype in the same pot. Pots were containing at least eight plants of each genotypes, $n = 10$ pots. Different letters represent significant differences between genotypes using a two-sided Mann–Whitney test ($p < 0.01$). centre lines show the medians; box limits indicate the 25th and 75th percentiles. **e** Boxplots showing the number of siliques produced per plants. Plants were cultivated in a high humidity environment for 10 days after germination and then transferred to a greenhouse. Each point represents the total number of seeds containing siliques per plant ($n = 12$ for WT, $n = 14$ for sgn3-3, $n = 12$ for myb36-2, $n = 11$ for sgn3-3 myb36-2, $n = 12$ for WT—pELTP::CDEF, $n = 15$ for sgn3-3 myb36-2—pELTP::CDEF). Different letters represent significant differences between genotypes using a two-sided Mann–Whitney test ($p < 0.01$). Centre lines show the medians; box limits indicate the 25th and 75th percentiles.

powerful tool for studying the role of endodermal barriers in a range of processes such as nutrient, hormone and water transport and biotic interaction with soil microorganisms[54].

The data presented here reveals that the Schengen-pathway is involved in the deposition of two chemically distinct types of lignin. The Schengen-pathway and MYB36 are required for the deposition of CS lignin. Constitutive activation of the Schengen-pathway leads to the deposition of a chemically distinct stress-like type of lignin. This deposition of stress-lignin contributes to sealing the apoplast and maintaining ion homoeostasis in the absence of CS integrity. However, no PM attachment to the CW is observed at the site of stress-lignin deposition as seen for the CS, suggesting an inferior seal is formed.

## Methods

**Plant material**. *A. thaliana* accession Columbia-0 (Col-0) was used for this study. The following mutants and transgenic lines were used in this study: sgn3-3 (SALK_043282)[9], myb36-2 (GK-543B11)[17], pCASP1::CASP1-GFP[12], ahp6-1[25], esb1-1[10], pELTP::CDEF[55], pELTP::SYP122-mCitrine.
The corresponding gene numbers are: SGN3, At4g20140; MYB36, At5g57620; CASP1, At2g36100; AHP6, At1g80100; ESB1, At2g28670; ELTP, At2g48140; CDEF, At4g30140; SYP122, At3g52400.

**Generation of transgenic lines**. The pELTP::mCit-SYP122 construct was obtained by recombining three previously generated entry clones for pELTP[56], mCITRINE and SYP122 cDNA[57] using LR clonase II (Invitrogen). This construct was independently transformed into WT, sgn3-3, myb36-2 or sgn3-3 myb36-2 using the floral dip method[58]. The construct pELTP::CDEF[55] was independently transformed into WT and sgn3-3 myb36-2.

**Growth conditions**. For agar plates assays, seeds were surface sterilized, sown on plates containing ½ Murashige and Skoog (MS) pH 5.8 with 0.8% agar, stratified for 2 days at 4 °C and grown vertically in growth chamber under long day condition (16 h light 100 µE 22 °C/8 h dark 19 °C) and observed after 6 days. The CIF2 peptide treatment (DY(SO3H)GHSSPKPKLVRPPFKLIPN) was applied from germination at a concentration of 100 nM. The CIF2 peptide was synthetized by Cambridge Peptided Ltd.
For ionomic analysis, plants were grown using three growth conditions:

- Sterile ½ MS agar plate. Seeds were surface sterilized and sown on plates containing ½ MS with 0.8% agar, stratified for 2 days at 4 °C and grown vertically in growth chamber under long day condition (16 h light 100 µE 22 °C/8 h dark 19 °C). Shoots were collected 2 weeks after germination.
- Hydroponic. Plants were grown for 5 weeks under short day condition (8 h light 100 µE 21 °C/16 h dark 18 °C) at 20 °C with a RH of 65% RH in a media at pH 5.7 containing 250 µM CaCl₂, 1 mM KH₂PO₄, 50 µM KCl, 250 µM K₂SO₄, 1 mM MgSO₄, 100 µM NaFe-EDTA, 2 mM NH₄NO₃, 30 µM H₃BO₃, 5 µM MnSO₄, 1 µM ZnSO₄, 1 µM CuSO₄, 0.7 µM NaMoO₄, 1 µM NiSO₄. Media was changed weekly.
- Natural soil. Plants were grown for 9 weeks in a growth chamber under short day condition (8 h light 100 µE 19 °C/16 h dark 17 °C) at 18 °C with a RH of

70% in a soil collected in the Sutton Bonington campus of the university of Nottingham (GPS coordinate: 52°49′59.7″N 1°14′56.2″W).

**Fluorescence microscopy**. To determine the functionality of the endodermal apoplastic barriers we used the apoplastic tracer PI (Invitrogen). PI is a fluorescent molecule which diffusion into the tissue layers of the root is blocked only after the dye reaches the differentiated endodermis[23]. Six-day-old Col-0 seedlings were incubated in a fresh solution of 10 µg/mL PI (prepared from a stock solution 1 mg/mL) for 10 min in the dark and then rinsed twice with water. Seedlings were carefully placed on a microscope slide with water and covered with a coverslip. Using a fluorescence microscope Leica CTR5000, ×20 magnification, we quantified the number of cells from the onset of elongation until the endodermal cells blocked the PI penetration to the stele.
For lignin staining with basic fuchsin, CASP1-GFP visualization and Calcofluor white M2R staining, 6-day-old roots were fixed in 4% paraformaldehyde and then washed twice for 1 min with 1x PBS and transferred in ClearSee solution[59] (10% xylitol, 15% Sodium deoxycholate, 25% urea) for 24 h. Then, the seedlings were stained overnight in 0.2% Basic Fuchsin in ClearSee for lignin staining. Basic Fuchsin solution was removed and the seedlings were washed three times for 120 min with ClearSee with gentle shaking. Then, the seedlings were stained for 1 h in 0.1% Calcofluor white M2R in ClearSee for CW staining. Calcofluor white M2R solution was removed and the seedlings were washed three times for 30 min with ClearSee with gentle shaking. Roots were carefully placed on a microscope slide with ClearSee and covered with a coverslip. The roots were then observed with confocal microscopes (Zeiss LSM500 and Leica SP8) using their respective software (Zeiss ZEN 2.3 and Leica LAS X). We used an excitation at 594 nm and an emission band path of 600–650 nm for Basic Fuchsin, 405 nm and an emission band path of 425–475 nm for Calcofluor white M2R and 488 nm and an emission band path of 505–530 nm for GFP.
Suberin staining was performed using Fluorol yellow 088[7,56]. Seedlings were incubated in a freshly prepared solution of Fluorol Yellow 088 (0.01%w/v, in lactic acid) at 70 °C for 30 min, followed by two rinses with water. Then, samples were treated with aniline blue (0.5% w/v, in water) at room temperature for 30 min in darkness. After two rinses with water, several roots were placed on microscope slides with water, covered with a coverslip and the suberin deposition pattern was quantified using an epifluorescence microscope Leica CTR5000, ×20 magnification with a GFP filter. The suberin deposition pattern was quantified as the number of cells in the continuous, patchy and no-suberin zones.

**Plasmolysis**. Plasmolysis was induced by mounting 6-day-old seedlings in 0.8 M mannitol on microscope slides and directly observed using confocal microscopy (Leica SP8). PA was used to inhibit lignin biosynthesis[7]. PA at a concentration of 10 µM was added to the media from germination. The proportion of the CW length in direct contact with the PM marker SYP122-mCitrine after plasmolysis was measured using Fiji after plasmolysis. This measurement was done on a maximum projection of the top endodermal cells as seen on Supplementary Fig. 6. The quantification represents the percentage of CW length in direct contact with the PM marker SYP122-mCitrine after plasmolysis. Plasmolysis events were imaged and quantified at 15 cells after the onset of elongation.
For the observation of CASP1-GFP and Lignin staining with basic fuchsin, the seedlings were incubated in 0.8 M mannitol for 5 min, and then fixed and cleared as described above.

**Thioacidolysis**. The plants were grown for 6 days on ½ MS plates supplemented with 10 nM 6-Benzylaminopurine and 0.1 % sucrose. Seeds were sown in three parallel lines per square plates (12 × 12 cm) at high density. Six plates were combined to obtain one replicate. The first 3 mm of root tips as this zone contains no xylem pole were collected in order to obtain 7–15 mg of dry weight. The samples were washed twice with 1 mL methanol, rotated for 30 min on a carousel and centrifugated to eliminate the methanol supernatant. This washing step was repeated once and the final methanol-extracted samples were then dried for 2 days at 40 °C (oven) before thioacidolysis.

The thioacidolyses were carried out in a glass tube with a Teflon-lined screwcap, from about 5 mg sample (weighted at the nearest 0.01 mg) put together with 0.01 mg C21 and 0.01 mg C19 internal standard (50 µL of a 0.2 mg/ml solution) and with 2 ml freshly prepared thioacidolysis reagent. The tightly closed tubes were then heated at 100 °C for 4 h (oil bath), with gentle occasional shaking. After cooling and in each tube, 2 ml of aqueous NaHCO3 0.2 M solution were added (to destroy the excess of BF3) and then 0.1 ml HCl 6 M (to ensure that the pH is acidic before extraction). The reaction medium was extracted with 2 ml methylene chloride (in the tube) and the lower phase was collected (Pasteur pipette) and dried over $Na_2SO_4$ before evaporation of the solvent (rotoevaporator). The final sample was redissolved in about 2 mL of methylene chloride and 15 µL of this solution were trimethylsilylated (TMS) with 50 µl BSTFA + 5 µl pyridine. The TMS solution was injected (1 µL) into a GC-MS Varian 4000 instrument fitted with an Agilent combiPAL autosampler, a splitless injector (280 °C), and an ion-trap mass spectrometer (electron impact mode, 70 eV), with a source at 220 °C, a transfer line at 280 °C and an $m/z$ 50–800 scanning range. The GC column was a Supelco SPB1 column (30 m × 0.25 mm i.d., film thickness 0.25 µm) working in the temperature programme mode from 45 to 160 °C at +30 °C/min and then 160–260 °C at +5 °C/min, with helium as the carrier gas (1 mL/min). The GC-MS determinations of the H, G and S lignin-derived monomers were carried out on ion chromatograms, respectively, reconstructed at $m/z$ 239, 269 and 299, as compared to the internal standard hydrocarbon evaluated on the ion chromatogram reconstructed at $m/z$ (57 + 71 + 85). Each genotype was analyzed as biological triplicates and each biological triplicate was subjected to two different silylations and GC-MS analyses.

**Raman microscopy**. Six-day-old seedling was fixed in PBS buffer containing 4% formaldehyde and 1% glutaraldehyde at 4°C overnight, then washed twice with PBS 30 min. Samples were progressively dehydrated in ethanol (30, 50, 70, 100% ethanol). Samples were aligned and embedded in Leica historesin using the protocol described in[60]. Sections of 5 µm at 4 mm from the root tip were generated using a Leica microtome.

The different samples were embedded in resin and cut at 4 mm from the root tip using a microtome with a thickness of 15 µm. Sections were mounted on superfrost glass slides. The samples were then mapped in a grid over the region of interest. The Raman imaging was performed with a Horiba LabRAM HR spectroscope equipped with a piezoelectric scan stage (Horiba Scientific, UK) using a 532 nm laser, a ×100 air objective (Nikon, NA = 0.9) and 600 g mm$^{-1}$ grating. Imaging was performed using the LabSpec 6.4.3 Spectroscopy Suite Software. Maps were collected for the regions of interest by setting equidistant points along the sample to ensure maximum coverage. The main regions covered in the analysis were the endodermal cell–cell junction, the endodermal cell corners towards the endodermal–cortical junction and the xylem poles (Supplementary Fig. 1). The maps were acquired with two accumulations and 30 s integration time. The spectra were acquired in the range 300–3100 cm$^{-1}$. The spectra were processed using MATLAB and eigenvector software. First, the spectra were trimmed (500–1800 cm$^{-1}$), smoothed and then baseline corrected using an automatic least squares algorithm. This was followed by a percentile mean subtraction (10–20%) to remove signal from the resin. Finally, Gaussian image smoothing was performed to improve signal to noise of the Raman maps. MCR analysis was performed on the maps containing the specific ROIs and the corresponding lignin spectra were extracted. These lignin spectra were then used as bounds for the MCR analysis of the large maps where the concentration of these spectra is determined, with a high intensity indicating a high concentration of the specific lignin (Fig. 2c).

**Monolignol feeding experiment**. The plants were grown vertically for 4 days on control condition (½ MS, 0.8% agar) in growth chamber under long day condition (16 h light 100 µE 22 °C/8 h dark 19 °C). Seedlings were then transferred for 3 more days on the same condition on the same media supplemented or not with 10 µM PA (Sigma-Aldrich, P49805), 10 µM PA with 20 µM *p*-coumaryl alcohol (Sigma-Aldrich, PHL82506), 10 µM PA with 20 µM coniferyl alcohol (Sigma-Aldrich, 223735), 10 µM PA with 20 µM sinapyl alcohol (Sigma-Aldrich, 404586) or 10 µM PA with 10 µM *p*-coumaryl alcohol and coniferyl alcohol. Apoplastic permeability was determined using PI and by quantifying the number of cells from the onset of elongation until the endodermal cells blocked the PI penetration to the stele.

For lignin staining, the roots of *myb36-2* were fixed with 4% PFA for 120 min at 20 °C. Seedlings were then washed twice for 1 min with 1x PBS and transferred to the Clearsee solution[59] (10% xylitol, 15% Sodium deoxycholate, 25% urea) for 24 h. Then, the seedlings were stained overnight in 0.2% Basic Fuchsin in ClearSee for lignin staining. Basic Fuchsin solution was removed and the seedlings were washed three times for 120 min with ClearSee with gentle shaking. Roots were carefully placed on a microscope slide with ClearSee and covered with a coverslip. We imaged the root at 20 cells after the onset of elongation using a confocal microscope Leica SP8, ×63 objective (NA = 1.2) by performing a z-stack on the top endodermal cells. We used an excitation at 594 nm and an emission band path of 600–650 nm for Basic Fuchsin. For the quantification of pixel intensity of basic fuchsin fluorescence, we performed a maximum intensity projection of the top endodermal cells. Using Fiji, we traced a 1 µm thick segmented line following the cortex-facing CW corner of one endodermal cell. The mean pixel intensity was determined along that line. This represents the pixel intensity for one endodermal cell. Pixel intensity was measured in three to six individual cells per plant. A total of four to six individual plants were measured for each treatment. Pixel intensities were plotted using the SuperPlots tool[61].

**RNA-seq**. The plants were grown for 6 days on ½ M/S plates. Seeds were sown in three parallel lines per square plates (12 × 12 cm) at high density. The first 5 mm of root tips were collected. One plate was used as a biological replicate. The samples were snap-frozen at harvest and ground into fine powder in a 2 mL centrifuge tube. Total RNA was extracted according to ref. [62]. Samples were homogenized in 400 µL of Z6-buffer containing 8 M guanidine-HCl, 20 mM MES, 20 mM EDTA pH 7.0. After the addition of 400 µl phenol:chloroform:isoamyl alcohol, 25:24:1, samples were vortexed and centrifuged (15,000 × g 10 min) for phase separation. The aqueous phase was transferred to a new 1.5 mL tube and 0.05 volumes of 1 N acetic acid and 0.7 volumes 96% ethanol was added. The RNA was precipitated at −20 °C overnight. Following centrifugation (15,000 × g 10 min, 4 °C), the pellet was washed with 200 µL 3 M sodium acetate at pH 5.2 and 70% ethanol. The RNA was dried and dissolved in 30 µL of ultrapure water and store at −80 °C until use. DNase treatment (DNase I, Amplification Grade, 18068015, Invitrogen) was carried out on the samples to remove genomic DNA. The RNA Concentration and quality were determined using Qubit (Invitrogen; Q10210) and TapeStation (Agilent; G2991A) protocols. Libraries were generated using the Lexogen Quant Seq 3′ mRNA Seq (FWD) Library Prep Kit (Lexogen; 015) which employs polyA selection to enrich for mRNA. Library yield was measured by Qubit (Invitrogen; Q10210) and TapeStation (Agilent; G2991A) systems using protocols to determine concentration and library size, these are then pooled together in equimolar concentrations. The concentration of the pool of libraries was confirmed using the Qubit and qPCR and then loaded onto an Illlumina NextSeq 500/550 High Output Kit v2.5 (75 Cycles) (Illumina; 20024906), to generate ~5 million 75 bp single-end reads per sample.

BBduk v38.82 was used to identify and discard reads containing the Illumina adaptor sequence[63]. Then, we mapped the resulting high-quality filtered reads against the TAIR10 Arabidopsis reference genome (Ensembl Plants v48) using STAR 2.7.5c[64] with the following parameters:

- outFilterType BySJout
- outFilterMultimapNmax 20
- alignSJoverhangMin 8
- alignSJDBoverhangMin 1
- sjdbGTFtagExonParentGene gene_id
- sjdbGTFfile Arabidopsis_thaliana.TAIR10.48.gtf
- outReadsUnmapped Fastx
- quantMode GeneCounts
- outFilterMismatchNmax 999 --outFilterMismatchNoverLmax 0.1 --alignIntronMin 20
- alignIntronMax 1000000 --alignMatesGapMax 1000000 --outSAMattributes NH HI NM MD
- outSAMtype BAM SortedByCoordinate
- outFileNamePrefix <outfile>

We used the R package DESeq2 v.1.24.0 to identify DEGs between each genotype, *sgn3-3*, *sgn3-3 myb36-2*, *sgn3-3* (+CIF2), *esb1-1*, *esb1-1 sgn3-3* and WT (+CIF2) against WT (Col-0). To do so, we fitted the following generalized linear model:

$$\text{Gene abundance} \sim \text{Rep} + \text{Genotype}$$

A gene was considered statistically differentially expressed if it had a false discovery rate (FDR) adjusted *p* value < 0.05.

For visualization purposes, we created a standardized gene matrix. To do so, we applied a variance stabilizing transformation to the raw count gene matrix followed up by standardizing the expression of each gene along the samples. We used this standardized gene matrix to perform principal coordinate (PC) analysis using the prcomp function in R. We displayed the results of the PC analysis using ggplot2.

Additionally, we subset the 3564 statistically significant DEGs from the standardized gene matrix. Then, for each DEG we calculated its mean expression across each genotype followed up by hierarchical clustering (R function hclust method ward.D2) using the euclidean distance for the genotypes and the correlation dissimilarity for the genes. To define the seven clusters of cohesively expressed genes, we cut the gene dendrogram from the hierarchical clustering using the R function cutree. We visualized the expression of the 3564 DEGs and the result of the clustering approach using ggplot2. We used the compareCluster function from the clusterProfiler R package to perform gene ontology (GO) analysis for the seven clusters of cohesively expressed DEGs.

We constructed individual heatmaps for the phenylpropanoid pathway, the peroxidases, the laccases and the aquaporin genes by sub setting the corresponding curated gene ids from the standardized gene matrix and procedure described above.

Raw sequence data and read counts are available at the NCBI Gene Expression Omnibus accession number (GEO: GSE158809). Additionally, the scripts created to analyze the RNA-Seq data can be found at https://github.com/isaisg/schengenlignin with an assigned DOI (https://doi.org/10.5281/zenodo.4588023).

**Split-root experiment**. Wild-type plants were grown 8 days vertically on ½ M/S square plates (12 × 12 cm) in a growth chamber under long day condition (16 h light 100 µE 22 °C/8 h dark 19 °C). The primary root was then cut below the first two lateral roots and plants were allowed to grow on the same plate for an additional 4 days. Then, plants were transferred on to a round plate (Ø: 10 cm) containing two compartments generated by a dividing partition. The root system of each plant was grouped into two halves and placed on either side of the partition. Two plants per plate were transferred. The compartments were filled with either solid ½ M/S on both compartments, solid ½ M/S supplemented with 100 nM CIF2 in both compartments or solid ½ M/S on one side and solid ½ M/S supplemented with 100 nM CIF2 on the other side. Plants were then grown for an additional 3 days. The roots from plants presenting balanced roots on both sides of the divide were harvested for lignin staining and RNA extraction. For RNA extraction, roots from four to six plates were combined to obtain one replicate and stored in liquid nitrogen. RNA extraction was performed as described in the "RNA-seq" section. 1 µg of total RNA treated with DNase I (Thermo Scientific) was used for reverse transcription (RevertAid First Strand CDNA synthesis kit; Thermo Scientific) with oligo(dT)18. cDNA was diluted twice with water, and 1 µL of each sample was assayed three time by qRT-PCR in a LightCycler 480 (Roche) using LC480-SYBR-Green master I (Roche). Expression levels were calculated relatively to the gene *At4g24550*[65] using the comparative threshold cycle method. The list of genes related to defence was determined by selecting the genes from the cluster C1 (Fig. 3) displaying an average log2 fold change in *myb36-2*, *esb1-1* and WT (+CIF2) higher than 2.9 and belonging to the following using the Gene Ontology annotations: defence response, defence response to bacterium, defence response to fungus, immune system process, innate immune response, response to chitin, response to fungus and immune response. All primer sets are indicated in Supplementary Data 4.

For lignin staining, roots were fixed in paraformaldehyde and cleared in ClearSee solution[59] for 24 h. Then, roots were stained overnight in 0.2% Basic Fuchsin in ClearSee for lignin staining. Basic Fuchsin solution was removed and replaced by 0.1% direct yellow 96 (CW) in ClearSee for 1 h. The roots were washed three times for 120 min with ClearSee with gentle shaking. Roots were carefully placed on a microscope slide with ClearSee and covered with a coverslip. We imaged the root at 15 cells after the onset of elongation using a confocal microscope Leica SP8, ×63 objective (NA = 1.2). We used an excitation at 594 nm and an emission band path of 600–650 nm for Basic Fuchsin and 488 nm and an emission band path of 500–540 nm for Direct yellow 96.

**Extraction and profiling of metabolites**. The plants (WT, *esb1-1*, *sgn3-3* and *esb1-1 sgn3-3*) were grown for 6 days on ½ M/S plates supplemented with 0.1% sucrose. Seeds were sown in three parallel lines per square plates (12 × 12 cm) at high density. The first 5 mm of root tips were collected in order to obtain 10–20 mg of dry weight per replicate. Eight plates were combined to obtain one replicate. Eight replicates per genotypes were harvested. The samples were snap-frozen at harvest and ground into fine powder in a 2 mL centrifuge tube then homogenized in liquid nitrogen and extracted with 1 ml methanol. The methanol extract was then evaporated, and the pellet dissolved in 200 µl water/cyclohexane (1/1, v/v). 10 µl of the aqueous phase was analyzed via reverse phase UHPLC (Acquity UPLC Class 1 systems consisting of a Sample Manager-FTN, a Binary Solvent Manager and a Column Manager, Waters Corporation, Milford, MA) coupled to negative ion ElectroSpray Ionization-Quadrupole-Time-of-Flight Mass Spectrometry (Vion IMS QTof, Waters Corporation) using an Acquity UPLC BEH C18 column (1.7 µm, 2.1 × 150 mm; Waters Corporation). Using a flow rate of 350 µl/min and a column temperature of 40 °C, a linear gradient was run from 99% aqueous formic acid (0.1%, buffer A) to 50% acetonitrile (0.1% formic acid, buffer B) in 30 min, followed by a further increase to 70% and then to 100% buffer B in 5 and 2 min, respectively. Full MS spectra (*m/z* 50–1500) were recorded at a scan rate of 10 Hz. The following ESI parameters were used: capillary voltage 2.5 kV, desolvation temperature 550 °C, source temperature 120 °C, desolvation gas 800 L/h and cone gas 50 L/h. Lock correction was applied. In addition to full MS analysis, a pooled sample was subjected to data-dependent MS/MS analysis (DDA) using the same separation conditions as above. DDA was performed between *m/z* 50 and 1200 at a scan rate of 5 Hz and MS → MS/MS transition collision energy of 6 eV. The collision energy was ramped from 15 to 35 eV and from 35 to 70 eV for the low and high mass precursor ions, respectively.

Integration and alignment of the *m/z* features were performed via Progenesis QI software version 2.1 (Waters Corporation). The raw data were imported in this software using a filter strength of 1. A reference chromatogram was manually chosen for the alignment procedure and additional vectors were added in chromatogram regions that were not well aligned. Peak picking was based on all

runs with a sensitivity set on "automatic" (value = 5). The normalization was set on "external standards" and was based on the dry weight of the samples[66]. In total, 13,091 *m/z* features were integrated and aligned across all chromatograms. Structural annotation was performed using a retention time window of 1 min, and using both precursor ion and MS/MS identity searches. The precursor ion search (10 ppm tolerance) was based on a compound database constructed via instant JChem (ChemAxon, Budapest, Hungary), whereas MS/MS identities were obtained by matching against an in-house mass spectral database (200 ppm fragment tolerance).

Using R vs 3.4.2., *m/z* features representing the same compound were grouped following the algorithm in[67]. Of the 13,091 *m/z* features, 12,326 were combined into 2482 *m/z* feature groups, whereas 765 remained as *m/z* feature singlets (i.e., low abundant features). All statistical analyses were performed in R vs. 3.4.2[68]. Including all *m/z* features and upon applying a prior inverted hyperbolic sine transformation[69], the data were analyzed via both PCA and one-way analysis of variance (ANOVA; lm() function) followed by Tukey Honestly Significant Difference (TukeyHSD() function) post hoc tests. For PCA, the R packages FactoMineR[70] and factoextra (https://CRAN.R-project.org/package=factoextra) were employed: PCA(scale.unit=T,graph=F), fviz_pca_ind() and fviz_pca_biplot (). Following ANOVA analysis, experiment-wide significant models were revealed via a FDR correction using the p.adjust(method = "fdr") function. Using a FDR-based *Q* value < 0.05, 4244 of the 13,091 *m/z* features were significantly changed in abundance corresponding to 123 *m/z* feature singlets and 1158 of the 2482 compounds. Using a minimum abundance threshold of 500 in at least one of the lines, further analysis was performed on 411 of the 1158 compounds and 11 of the 123 *m/z* feature singlets (411 compounds and 11 singlets representing together 889 *m/z* features).

**Root hydraulic conductivity**. The procedure was exactly identical to the one described in[48]. Root hydrostatic conductance (*K*r) was determined in freshly detopped roots using a set of pressure chambers filled with hydroponic culture medium. Excised roots were sealed using dental paste (Coltène/Whaledent s.a.r.l., France) and were subjected to 350 kPa for 10 min to achieve flow stabilization, followed by successive measurements of the flow from the hypocotyl at pressures 320, 160 and 240 kPa. Root hydrostatic conductance (*K*r) was calculated by the slope of the flow (*J*v) to pressure relationship. The hydrostatic water conductivity (*Lp*$_{r-h}$, ml H2O g$^{-1}$ h$^{-1}$ MPa$^{-1}$) was calculated by dividing *K*r by the root dry weight.

**Humidity, leaf surface, mortality, ionome and fitness**. For the determination of the leaf surface and mortality, the seeds were stratified for 2 days at 4 °C and the plants were grown in Levington M3 compost in a growth chamber under long day condition (16 h light 100 µE 21 °C/8 h dark 19 °C). The plants were grown for 7 days with high RH (80% RH) and then half of the plants were transferred at a lower humidity (60% RH). Leaf surface was determined at 6, 9, 12 and 15 days after germination using the threshold command of the FiJi software. The plants displaying no growth after 9 days and showing necrosis in all the leave surface were considered as dead plants.

For the determination of the shoot ionome, the plants were cultivated at 80% RH for 10 days and then transferred at 80% RH or 60% RH for 5 additional days. Shoots were harvested for ionomic analysis.

For the determination of the siliques number, the plants were cultivated in a high humidity environment for 10 days after germination and then transferred to a greenhouse. After siliques ripening, only the seeds containing siliques were counted.

**Ionomic analysis with ICP-MS**. Ionomics analysis of plants grown in soil (or on plate, hydroponically) was performed as described[71]. Briefly, samples (shoot) were harvested into Pyrex test tubes (16 × 100 mm) and dried at 88 °C for 20 h. After weighing the appropriate number of samples (these masses were used to calculate the rest of the sample masses; alternatively, all samples were weighed individually— usually for small set of samples), the trace metal grade nitric acid Primar Plus (Fisher Chemicals) spiked with indium internal standard was added to the tubes (1 mL per tube). The samples were then digested in dry block heater (DigiPREP MS, SCP Science; QMX Laboratories, Essex, UK) at 115 °C for 4 h. The digested samples were diluted to 10 mL with 18.2 MΩcm Milli-Q Direct water (Merck Millipore). Elemental analysis was performed with an ICP-MS, PerkinElmer NexION 2000 equipped with Elemental Scientific Inc. autosampler, in the collision mode (He) and Syngistix software. Twenty elements (Li, B, Na, Mg, P, S, K, Ca, Mn, Fe, Co, Ni, Cu, Zn, As, Se, Rb, Sr, Mo and Cd) were monitored. Liquid reference material composed of pooled samples was prepared before the beginning of sample run and was used throughout the whole samples run. It was run after every ninth sample to correct for variation within ICP-MS analysis run[71]. The calibration standards (with indium internal standard and blanks) were prepared from single element standards (Inorganic Ventures; Essex Scientific Laboratory Supplies Ltd, Essex, UK) solutions. Sample concentrations were calculated using external calibration method within the instrument software. Further data processing was performed using Microsoft Excel spreadsheet.

## Data availability
The data that support the findings of this study are available within the paper and its Supplementary Information or are available from the corresponding author upon reasonable request. The source data underlying Figs. 3 and 6b, Supplementary Fig. 3 and Supplementary data 1 are available at the NCBI Gene Expression Omnibus accession number (GEO: GSE158809). Source data are provided with this paper.

## Code availability
All scripts used for the RNA-seq analysis are available on GitHub (https://github.com/isaisg/schengenlignin) with an assigned DOI (https://doi.org/10.5281/zenodo.4588023).

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

## Acknowledgements

We thank Deep Seq (Next Generation Sequencing Facility of the University of Nottingham, UK), the nmRC (Nanoscale and Microscale Research Centre of the University of Nottingham, UK), the Microscopy and Histology Facility of the University of Aberdeen (UK), the VIB Metabolomics Core (VIB-UGent, Belgium). This work was supported by grants from the UK Biotechnology and Biological Sciences Research Council Grant (Grant Nos. BB/L027739/1 and BB/N023927/1 to D.E.S.), the Coordinating Action in Plant Sciences Promoting sustainable collaboration in plant sciences (Grant Nos. ERACAPS13.089_RootBarriers to D.E.S.), the Engineering and Physical Sciences Research Council (Grant No. EP/R025282/1) and the Future Food Beacon of Excellence at the University of Nottingham (Nottingham Research Fellowship to G.C. and Postdoctoral Research Fellowship to G.R.).

## Author contributions

G.R., P.R., G.C. and D.E.S. conceived and designed the experiments, I.S.-G. analyzed RNA-seq data, S.F. generated the endodermal plasma membrane marker, G.R. and P.R. performed Raman imaging, A.L., D.T. and M.W.G. analyzed Raman data, C.L. performed thioacidolysis experiments, K.M. analyzed metabolomic data, M.C.P. and Y.B. performed hydraulic conductivity measurements, P.F. performed ICP-MS measurements, G.R. wrote the paper with multiple rounds of review and editing from D.E.S., G.R., S.F., C.L., P.F., N.G., Y.B., W.B., M.W.G., G.C. and D.E.S. All contributed to final editing and review.

## Competing interests

The authors declare no competing interests.
