## [Peer Review File · Nature Communications]

REVIEWER COMMENTS

Reviewer #1 (Remarks to the Author):

This is overall an excellent manuscript that addresses an area of increasing interest and importance in plant biology. The past few years have seen major advances in our understanding of the composition of the root endodermis and Casparian strip (CS), and the CS has emerged as a model for understanding the spatial control of lignification in plant cell walls. The present manuscript adds to our understanding of these processes, while also providing additional information on the functions of the endodermis and CS in plant water and solute relations.

The methodology used appears to be sound and explained in sufficient detail to allow for replication, and this work clearly makes an important new contribution. I do not see any major flaws, but the authors might consider the comments below.

The authors provide convincing genetic and cytological evidence for the operation of the Schengen pathway in controlling the formation of a "defense" type lignin that can compensate for barrier defects. There are, however, several questions that occurred to me which could have been addressed, either experimentally (although maybe this is too much to ask) or at least through discussion.

1. What is the significance of the compositional change (increased H-units) in the Schengen-controlled lignin deposited in the cell corners? This lignin clearly has a different composition from the bulk lignin, but it was my understanding that the lignin deposited in cell corners early during the lignification process is usually somewhat H-rich. The authors argue that this is more like defense lignin, but in some plants this tends to be more S-rich. Have the authors considered a strategy to induce this lignin with the same spatial distribution but with more "normal" composition (e.g. by overexpression of C3'H?) and test whether there are any functional consequences?

2. Activation of the Schengen pathway indeed appears to induce many of the genes of lignification, but not the C3'H necessary for moving from a hydroxyphenyl to a caffeyl/coniferyl moiety. But what about the fact that HCT, the first enzyme of the shikimate shunt to the caffeyl moiety, appears to be expressed normally. And can the authors comment on the expression of laccases and peroxidases in relation to Schengen control, as the distinction between the roles of these two classes of enzymes has been a major feature of the recent findings on the mechanism of lignification in the CS.

3. The activation of defense response genes in addition to abnormal lignification is interesting, but it would have been helpful to know exactly where the major induced defense gene products are localized within the root. Are any found in the cell wall (some PR proteins become wall-localized and where they can generate secondary defense signals). Is the response local or systemic?

One structural weakness of the paper is that, after the discussion of effects of CS modification on cell wall attachment, the paper seems, to this reviewer at least, to move onto a different topic. The same mutants and treatments are used, but now from the perspective of the function of the CS in solute and water relations. The science is still very good and interesting, but the connection with the first part of the manuscript could be made better. It almost feels as though two separate stories have been sewn together, particularly since the title of the paper does not refer to the results in "part two".

Minor points:

Line 299. This sentence is not clear as to whether the present results are confirming or contradicting previous reports.

Figure S4. I could not see the asterisks

In several places in the manuscript, (e.g. L236, 75-76, 55) the verb does not correspond to the

noun that serves as the subject to the phrase (e.g. large amounts of lignin is deposited....).

Reviewer #2 (Remarks to the Author):

The manuscript by Reyt et al. described that the Shengen-pathway controls deposition of compensatory lignin at the cell corners of the endodermal cells. The chemical composition of the lignin was characterized to be distinct from the CS lignin but similar to the stress- and pathogen response lignin. The RNA-seq analysis revealed that cell corner lignification is preceded by an induction of the phenylpropanoid pathway and an inactivation of aquaporin expression. The authors also characterized nutrients homeostasis and water balance in previously identified mutants and a *sgn3 myb36* double mutant which completely lack endodermal lignin. The results identified elements with a strong positive correlation between an increased CS permeability and those with a strong negative correlation. Finally the authors showed that the mutant completely lack endodermal lignin is sensitive to low humidity and the cell corner lignin significantly contribute to survival in low humidity. I acknowledge that these findings are original and novel, and significantly improve our understanding of signaling of endodermal barrier formation and its physiological significance. The establishment of the mutants completely lack endodermal lignin and also suberin will lead to further understand the physiological significance of the endodermal barriers. The experiments are well designed and conducted, and the manuscript is clearly written.

Only one concern I have is regarding the significance of the cell corner lignin in normal plants. Is the cell corner lignin established when the Casparian strip is defective by a kind of stress? Does the phenomenon occur only in *myb36* or *ebs1* mutants? Please discuss the significance in normal plants.

Reviewer #3 (Remarks to the Author):

The manuscript by Reyt et al., investigates how Arabidopsis seedling roots respond to impairment of the mechanisms modulating Casparian strip formation, confirms that the Casparian strip has an essential role in controlling ion uptake from soil and highlights how the controlled manipulation of Casparian formation can be used to study the plant processes responsible for ion uptake from the soil. Initially they determined that *SGN3* and *MYB36* jointly regulate CS lignification while only *SGN3* is essential for the "compensatory" lignification in the cell corners. They used a combination of genetic experiments and chemical analysis to show that the "corner lignin" is distinct from wildtype CS and vascular strand lignin. They proceed to investigate how the Schengen pathway regulates phenyl-propanoid biosynthesis, identify a specific pathway element which could be regulated and highlight how similarities between the Schengen and the MAMP pathways with respect to control of defense compound production. They continue by investigating the effects of CS manipulation on plasma membrane attachment and show that the CS domain is relevant and not the lignin. Based on the effects observed regarding the plasma membrane cell wall attachments they investigate how manipulating CS formation affects ion uptake from different environments and observe specific effects on certain ions (upon CS reduction: increased levels of Li, As, Mn, Na, Sr, S, Cu, Ca, B and reduced ones for Fe, Cd, P, Zn, K, Rb). They also investigated root hydraulic conductivity with their results suggesting that *MYB36* is required via possibly regulating aquaporin gene expression. They finish the manuscript by investigating the effects of manipulating endodermis root barrier formation in low/high humidity environments on plant growth and can show that the more severe the endodermal barrier is reduced the more profound plant fitness is affected. The last finding is on one hand rather expected / trivial but on the other hand creates interesting opportunities for future research.

As is illustrated by the length of the previous paragraph, the manuscript contains a lot of

experiments, which form a thorough investigation into the mechanisms regulating biosynthetic processes when CS formation is disturbed and what the consequences are on plant general health and fitness. These are topics of general interest, the experiments to address them are well-designed and I find the results presented overall concise, relevant and interesting. Summarized in a simplified manner the authors initially dissect quite elegantly the molecular mechanisms required. They proceed then to convincingly show that without a functional CS plants have real problems to survive and that studies in artificial laboratory conditions may not uncover the full impact loss of the CS has on plant growth and fitness. While the latter represent important findings, I also think they are somewhat obvious. The only reason why I think they should be retained, is the argument that roots manipulated in this manner could be possibly used as model system to study root/soil uptake processes.

Major point:

I wonder if the authors can comment on the following. I suspect the material for the ionic analysis derives from soil-grown plants, which experienced 80% RH. Based on the mortality data in Figure 6D, such plants exhibited no significantly increased mortality rate. That seems to raise the question to what extent the changes observed in ion distribution are actually relevant for plant survival. I think the authors should actually investigate what happens to the ion distribution in plants exposed to 60% RH / greenhouse conditions, where they observe the profound fitness reduction. I appreciate that this is a lot of work so perhaps they can alternatively measure fitness in plants exposed to 80% RH throughout their life, where they have already measured ion levels.

Minor points:

Line 55: are deposited not is

Line 164: is the composition really unique or simply different?

Line 195: Is it really a "novel" form of stress lignin? Is it really novel or simply found in a novel location/process?

Line 236: has been

Main figure 4 is labelled "supplemental"

Below we copy the reviewers comments and our detailed individual response (**in bold**)

Reviewer #1 (Remarks to the Author):

This is overall an excellent manuscript that addresses an area of increasing interest and importance in plant biology. The past few years have seen major advances in our understanding of the composition of the root endodermis and Casparian strip (CS), and the CS has emerged as a model for understanding the spatial control of lignification in plant cell walls. The present manuscript adds to our understanding of these processes, while also providing additional information on the functions of the endodermis and CS in plant water and solute relations.

The methodology used appears to be sound and explained in sufficient detail to allow for replication, and this work clearly makes an important new contribution. I do not see any major flaws, but the authors might consider the comments below.

The authors provide convincing genetic and cytological evidence for the operation of the Schengen pathway in controlling the formation of a “defense” type lignin that can compensate for barrier defects. There are, however, several questions that occurred to me which could have been addressed, either experimentally (although maybe this is too much to ask) or at least through discussion.

We thank the reviewer for their positive comments about the work.

1. What is the significance of the compositional change (increased H-units) in the Schengen-controlled lignin deposited in the cell corners? This lignin clearly has a different composition from the bulk lignin, but it was my understanding that the lignin deposited in cell corners early during the lignification process is usually somewhat H-rich. The authors argue that this is more like defense lignin, but in some plants this tends to be more S-rich. Have the authors considered a strategy to induce this lignin with the same spatial distribution but with more “normal” composition (e.g. by overexpression of C3'H?) and test whether there are any functional consequences?

Using an overexpressing line of C3'H could potentially lead to a more “normal” lignin composition. However, from the literature such a line (e.g. *ref8^{OpON}*) appears to still accumulate more H-unit than WT plants probably due to a partial complementation of C3'H (<https://doi.org/10.1104/pp.113.229393>). Further, it would be more challenging to investigate the functional relevance of cell corner lignin due to the presence of a functional Casparian strip (CS) in *ref8^{OpON}*. To test if the compositional change observed in response to the activation of the Schengen pathway has functional consequences, we instead used the *myb36-2* mutant displaying cell-corner lignin only and no CS lignin. Using a pharmacological approach, we blocked endogenous monolignol production with piperonylic acid (PA), an inhibitor of the phenylpropanoid pathway. We then fed the plants with each monolignol separately (20 μM *p*-coumaryl, 20μM coniferyl, or 20μM sinapyl alcohols) or with a combination of the two main monolignols incorporated in cell-corner lignin (10 μM *p*-coumaryl + 10 μM coniferyl alcohols). The application of *p*-coumaryl or coniferyl alcohols alone can trigger cell-corner lignification but not sinapyl alcohol. The combined application of *p*-coumaryl and coniferyl alcohols increased the deposition of cell corner lignin in comparison with the single addition of these monolignols. We then tested the capacity of this modified

cell-corner lignin to block the endodermal apoplastic pathway using propidium iodide. Feeding with coniferyl alcohol can partially recover the effect of PA on PI permeability. Feeding with *p*-coumaryl alcohol shows little effect on PI permeability, and sinapyl alcohol has no effect. Strikingly, the combined application of *p*-coumaryl and coniferyl alcohols can fully complement the inhibitor-induced defect in root permeability. These data indicate that each monolignol has a different capacity of incorporation into cell-corner lignin and different properties for sealing the endodermal apoplast. The addition of *p*-coumaryl alcohol with the main monolignol coniferyl alcohol seems to enhance lignin polymerisation and sealing of the apoplast. The improved sealing of the apoplast when both *p*-coumaryl alcohol and coniferyl alcohol are present in vitro supports the conclusion that increasing H-units in cell corner lignin in vivo should improve its ability to form an effective seal. We added these results to the main text (see paragraph: “Cell-corner lignin composition controls root permeability”, Line 199), in Supplementary Fig. 2 and in the M&M section.

H-rich lignin is known to be firstly deposited in cell corners early during the lignification process in pine (<https://doi.org/10.1007/BF00386021>). Defense lignin can also be S-rich lignin (<https://doi.org/10.1016/j.phytochem.2006.11.011>) and seems to provide enhance resistance to pathogens (<https://doi.org/10.1111/nph.15258>). Different compositional changes are observed according to plant species and biotic/abiotic stresses.

A recent report showed that H-rich lignin is accumulated in response to *Pseudomonas syringae* in *A. thaliana* (<https://doi.org/10.1111/nph.15258>). Interestingly, this enrichment partially relies on the transcription factor MYB15 controlling defence-induced lignification and basal immunity. The expression of MYB15 is also induced by the Schengen pathway. This was added in the text to emphasize similarities with defense lignin (Line: 265-267).

2. Activation of the Schengen pathway indeed appears to induce many of the genes of lignification, but not the C3'H necessary for moving from a hydroxyphenyl to a caffeyl/coniferyl moiety. But what about the fact that HCT, the first enzyme of the shikimate shunt to the caffeyl moiety, appears to be expressed normally. And can the authors comment on the expression of laccases and peroxidases in relation to Schengen control, as the distinction between the roles of these two classes of enzymes has been a major feature of the recent findings on the mechanism of lignification in the CS.

We included in the manuscript that HCT is not regulated by the Schengen pathway (Line 241-244). The activation of all the main enzymes of the phenylpropanoid pathway, apart from C3'H and HCT, we observed after triggering the Schengen pathway could explain the high level of H-units incorporation into endodermal cell-corner lignin.

Additionally, we added the expression of laccases and peroxidases in relation to the Schengen pathway. We observed a large set of peroxidases and laccases upregulated by the activation of the Schengen pathway. This set of genes shares a high similarity with the set of the peroxidases and laccases previously identified as upregulated in response to CIF2 in <https://doi.org/10.15252/embj.2019103894>. These results have been added in Supplementary Fig 3c, in the main text (Line: 237-241) and in the M&M section.

The recent findings on these two classes of enzymes showed that peroxidases but not laccases are required for CS lignification (<https://doi.org/10.1073/pnas.2012728117>).

The laccases LAC1, 3, 5 and 13 accumulate at the CS and in the endodermal cell corner. The expression of LAC1, 5, 12 and 13 is induced by the Schengen pathway (Supplementary Fig. 3c) suggesting their implication in cell-corner lignin deposition. However, this does not seem to be the case since exogenous CIF2 application triggers ectopic lignin deposition in the 9x *lac* mutant (*lac1;3;5;7;8;9;12;13;16*) (<https://doi.org/10.1073/pnas.2012728117>). The peroxidases PER3, 9, 39, 64 and 72 are required for CS lignin deposition as shown in the 5x *per* mutant (*per3;9;39;64;72*) fully lacking CS lignin. This mutant still exhibits ectopic lignification due to the constitutive activation of the Schengen pathway (<https://doi.org/10.1073/pnas.2012728117>). This indicates that other peroxidases induced by the Schengen pathway as found in Supplementary Fig. 3c could be responsible for cell-corner lignin deposition.

3. The activation of defense response genes in addition to abnormal lignification is interesting, but it would have been helpful to know exactly where the major induced defense gene products are localized within the root. Are any found in the cell wall (some PR proteins become wall-localized and where they can generate secondary defense signals). Is the response local or systemic?

Activation of the Schengen-pathway shared similarities with MAMP (microbe-associated molecular pattern) perception as highlighted in <https://doi.org/10.15252/embj.2019103894>. Some of the close homologs of SGN3 are PEPR1 and PEPR2 (<https://doi.org/10.1371/journal.pgen.1007847>) that are receptors to endogenous plant peptides (PEPs), whose activities resemble that of MAMP (<https://doi.org/10.1016/j.tplants.2017.07.005>). Receptor kinases such as FLS2 or EFR bind MAMPs and interact with SERK family of co-receptor (<https://doi.org/10.1038/nature05999>). SGN3 also interact with SERK proteins (<https://doi.org/10.1073/pnas.1911553117>). During MAMP perception, the signal is transduced through kinases of the RLCKVII family, such as BIK1 or PBLs, which are homologs of SGN1 (<https://doi.org/10.1146/annurev-arplant-042817-040540>). BIK1 and SGN1 were shown to induce RBOHD driving ROS burst (<https://doi.org/10.1016/j.chom.2014.02.009>, <https://doi.org/10.15252/embj.2019103894>). Kinases of the RLCKVII family directly phosphorylate mitogen-activated protein kinases (MAPKs, <https://doi.org/10.1105/tpc.17.00981>) and this is also the case during the activation of the Schengen pathway (<https://doi.org/10.15252/embj.2019103894>). Further, MYB15, a transcription factor shown to be involved in MAMP-induced lignification (<https://doi.org/10.1105/tpc.16.00954>) is also activated by the Schengen pathway. Therefore, this illustrates striking similarities of the Schengen-pathway with defense response.

To determine if the activation of the Schengen-pathway trigger local and/or systemic activation of defence-related genes, we used a split-root system, in which two equal portions of the root system of a single plant were physically separated with a barrier. One half of the root system was exposed to the Schengen-pathway activator CIF2, whereas the other side was kept in the same medium without CIF2 (Supplementary Fig. 5a). Plants were exposed to this split-root set up for 3 days. We observed that cell-corner lignification occurs only in roots in direct contact with CIF2 (Supplementary Fig. 5b) as observed for the induction of the expression of peroxidase genes likely involved in lignin polymerisation (Supplementary Fig. 5c). We then selected a list of 9 genes induced by the Schengen pathway (C1 of Fig. 3a) and related to defence based on

their Gene ontology annotation. We found that all the tested genes are exclusively induced locally in the presence of CIF2 (Supplementary Fig. 5d). Interestingly, the expression of most of the tested genes are also induced by immunity elicitors, the bacterial flagellin-derived flg22 (3 genes out of the 9 tested genes) and/or the endogenous Pep1 peptide (8 genes out of the 9 tested genes) (<https://doi.org/10.1105/tpc.20.00154>). These new results confirm similarities of the Schengen-pathway with defence response and show for the first time that activation of the Schengen-pathway occurs locally and not systemically. These new results are shown in Supplementary Fig. 5d, in the main text (Paragraph: “Local activation of genes related to defense.” line: 290-306) and in the M&M section.

One structural weakness of the paper is that, after the discussion of effects of CS modification on cell wall attachment, the paper seems, to this reviewer at least, to move onto a different topic. The same mutants and treatments are used, but now from the perspective of the function of the CS in solute and water relations. The science is still very good and interesting, but the connection with the first part of the manuscript could be made better. It almost feels as though two separate stories have been sewn together, particularly since the title of the paper does not refer to the results in “part two”.

The title has been modified to refer to the results of the second part of the manuscript. (New title: Two chemically distinct root lignin barriers control solute and water balance). We improved the transition after the discussion of the effects of endodermal lignin on cell wall attachment (Line 349-352).

Minor points:

Line 299. This sentence is not clear as to whether the present results are confirming or contradicting previous reports.

The present results are confirming previous reports. This sentence has been clarified and now it reads: “These findings confirms previous reports^{13,18} showing that CS lignin is not required for the formation of the CSD.” (Line: 340-342)

Figure S4. I could not see the asterisks

Asterisks have been added to the Figure.

In several places in the manuscript, (e.g. L236, 75-76, 55) the verb does not correspond to the noun that serves as the subject to the phrase (e.g. large amounts of lignin is deposited...).

We applied these corrections to the manuscript.

Reviewer #2 (Remarks to the Author):

The manuscript by Reyt et al. described that the Shengen-pathway controls deposition of compensatory lignin at the cell corners of the endodermal cells. The chemical composition of the lignin was characterized to be distinct from the CS lignin but similar to the stress- and pathogen response lignin. The RNA-seq analysis revealed that cell corner lignification is preceded by an induction of the pheylopanoid pathway and an inactivation of aquaporin

expression. The authors also characterized nutrients homeostasis and water balance in previously identified mutants and a *sgn3 myb36* double mutant which completely lack endodermal lignin. The results identified elements with a strong positive correlation between an increased CS permeability and those with a strong negative correlation. Finally the authors showed that the mutant completely lack endodermal lignin is sensitive to low humidity and the cell corner lignin significantly contribute to survival in low humidity. I acknowledge that these findings are original and novel, and significantly improve our understanding of signaling of endodermal barrier formation and its physiological significance. The establishment of the mutants completely lack endodermal lignin and also suberin will lead to further understand the physiological significance of the endodermal barriers. The experiments are well designed and conducted, and the manuscript is clearly written.

We thank the reviewer for their positive comments about the work.

Only one concern I have is regarding the significance of the cell corner lignin in normal plants. Is the cell corner lignin established when the Casparian strip is defective by a kind of stress? Does the phenomenon occur only in *myb36* or *ebs1* mutants? Please discuss the significance in normal plants.

This is an important point. Here, we described cell corner lignification in CS mutants (*myb36* and *esb1*). Activation of the Schengen pathway and subsequent cell corner lignification occurs when the integrity of the endodermal apoplastic barrier is lost. This can also occur during developmental processes such as lateral root emergence and during infection with soil-borne pathogens. Interestingly, an increased deposition of endodermal barriers and production of soluble phenolics have been shown to be associated with the resistance against a large range of soil-borne pathogens such as *Aphanomyces euteiches* (doi.org/10.1094/MPMI-22-9-1043), *Ralstonia solanacearum* (doi.org/10.1104/pp.109.141523) and *Phytophthora sojae* (doi.org/10.1104/pp.106.091090). We also recently showed that the plant microbiota can modulate the deposition of endodermal barriers (doi.org/10.1126/science.abd0695). We included this discussion in the manuscript (Line 302-306).

Reviewer #3 (Remarks to the Author):

The manuscript by Reyt et al., investigates how Arabidopsis seedling roots respond to impairment of the mechanisms modulating Casparian strip formation, confirms that the Casparian strip has an essential role in controlling ion uptake from soil and highlights how the controlled manipulation of Casparian formation can be used to study the plant processes responsible for ion uptake from the soil. Initially they determined that SGN3 and MYB36 jointly regulate CS lignification while only SGN3 is essential for the “compensatory” lignification in the cell corners. They used a combination of genetic experiments and chemical analysis to show that the “corner lignin” is distinct from wildtype CS and vascular strand lignin. They proceed to investigate how the Schengen pathway regulates phenylpropanoid biosynthesis, identify a specific pathway element which could be regulated and highlight how similarities between the Schengen and the MAMP pathways with respect to control of defense compound production. They continue by investigating the effects of CS manipulation on plasma membrane attachment and show that the CS domain is relevant and not the lignin. Based on the effects observed regarding the plasma membrane cell wall

attachments they investigate how manipulating CS formation affects ion uptake from different environments and observe specific effects on certain ions (upon CS reduction: increased levels of Li, As, Mn, Na, Sr, S, Cu, Ca, B and reduced ones for Fe, Cd, P, Zn, K, Rb). They also investigated root hydraulic conductivity with their results suggesting that MYB36 is required via possibly regulating aquaporin gene expression. They finish the manuscript by investigating the effects of manipulating endodermis root barrier formation in low/high humidity environments on plant growth and can show that the more severe the endodermal barrier is reduced the more profound plant fitness is affected. The last finding is on one hand rather expected / trivial but on the other hand creates interesting opportunities for future research.

As is illustrated by the length of the previous paragraph, the manuscript contains a lot of experiments, which form a thorough investigation into the mechanisms regulating biosynthetic processes when CS formation is disturbed and what the consequences are on plant general health and fitness. These are topics of general interest, the experiments to address them are well-designed and I find the results presented overall concise, relevant and interesting. Summarized in a simplified manner the authors initially dissect quite elegantly the molecular mechanisms required. They proceed then to convincingly show that without a functional CS plants have real problems to survive and that studies in artificial laboratory conditions may not uncover the full impact loss of the CS has on plant growth and fitness. While the latter represent important findings, I also think they are somewhat obvious. The only reason why I think they should be retained, is the argument that roots manipulated in this manner could be possibly used as model system to study root/soil uptake processes.

We thank the reviewer for their positive comments about the work. We agree with the reviewer that the mutant and lines generated in this study can be used as a model for studying the role of endodermal barriers in a range of processes such as nutrient, hormone and water transport and biotic interaction with soil microorganisms.

Authors from a previous study were surprised by the robustness of plant growth and the minimal impact on mineral nutrient homeostasis observed in a *sgn3* mutant (doi.org/10.7554/eLife.03115). This was explained in the paper by the partial nature of the apoplastic barrier defect observed in *sgn3*. The double mutant *sgn3 myb36* characterised in this study, completely lacks an apoplastic barrier, and as a result clearly demonstrates that the endodermal apoplastic barrier is critical for maintaining nutrient homeostatic and consequently growth and fitness.

Major point:

I wonder if the authors can comment on the following. I suspect the material for the ionic analysis derives from soil-grown plants, which experienced 80% RH. Based on the mortality data in Figure 6D, such plants exhibited no significantly increased mortality rate. That seems to raise the question to what extent the changes observed in ion distribution are actually relevant for plant survival. I think the authors should actually investigate what happens to the ion distribution in plants exposed to 60% RH / greenhouse conditions, where they observe the profound fitness reduction. I appreciate that this is a lot of work so perhaps they can alternatively measure fitness in plants exposed to 80% RH throughout their life, where they have already measured ion levels.

Analyses presenting the correlation between mineral nutrient and trace element accumulation and root permeability in Fig. 5b derives from plants grown in plates where the humidity is near saturation. In such conditions, the *sgn3-2 myb36-2* mutant did not show any growth defects, as seen in Supplementary Fig. 7. However, *sgn3-2 myb36-2* mutant displays a severe growth defect when grown in compost and exposed to lower humidity, as seen in Fig. 6. To determine if changes in mineral nutrient accumulation could explain this growth defect, we analysed the leaf ionomes of plants grown in compost and exposed to 60% RH and 80% RH. In WT, we observed that 60% RH triggers a significant decrease in the concentration of Cd, Zn, K (and chemical analogue Rb), and Ca (and chemical analogue Sr). In contrast, at 60% RH the *sgn3-2 myb36-2* double mutant does not show any significant decrease in element concentrations, and instead shows a significant increased accumulation of K (and its chemical analogue Rb) and Na (and its chemical analogue Li). This demonstrates that plants with no Casparian strips, or compensatory corner lignin, lose control of their nutrient balance in response to low humidity. Low humidity generates a higher transpiration rate, and consequently leads to a more uncontrolled and potentially detrimental accumulation of elements in the leaves, such as Na. This uncontrolled ion accumulation could be one component causing the strong growth defect observed in the absence of endodermal lignification. We added these new data to the main text (line: 444-458), Supplementary Fig. 8c and in the M&M section.

Minor points:

Line 55: are deposited not is

We corrected this.

Line 164: is the composition really unique or simply different?

Cell-corner lignin has a different chemical composition compared to both CS and xylem lignin. We replaced “unique” with “different”.

Line 195: Is it really a “novel” form of stress lignin? Is it really novel or simply found in a novel location/process?

We removed “novel” from the text.

Line 236: has been

Correction done

Main figure 4 is labelled “supplemental”

We corrected this mislabelling.

Additional modification:

The bar graph in Fig. 6a has been replaced by a boxplot.

Subheadings of the Results and Discussion section have been shortened to be less than 60 characters.

The RNAseq data have been reanalysed using a different aligner with more appropriate and optimized parameters specific for the type of RNA-Seq library used in this manuscript (Lexogen Quant Seq 3' mRNA Seq Library). We updated the figures containing genes expression data (Fig.3, Fig.6B and Sup. Fig. 3). These new results are very similar to the previous one and didn't change the interpretation or conclusions.

REVIEWERS' COMMENTS

Reviewer #1 (Remarks to the Author):

The authors have done an excellent job in responding to the comments I made, and I commend them for their effort in designing new experiments to "fill in the gaps". This is an excellent paper, and I recommend that it now be published.

Richard Dixon

Reviewer #2 (Remarks to the Author):

I see the concerns raised by reviewers were sufficiently addressed.

Reviewer #3 (Remarks to the Author):

The authors have addressed my concerns in a satisfactory manner.